# Locally anchoring enzymes to tissues via extracellular glycan recognition

Shaheen A. Farhadi[1], Evelyn Bracho-Sanchez[1], Margaret M. Fettis[1], Dillon T. Seroski[1], Sabrina L. Freeman[1], Antonietta Restuccia[1], Benjamin G. Keselowsky[1] & Gregory A. Hudalla [1]

Success of enzymes as drugs requires that they persist within target tissues over therapeutically effective time frames. Here we report a general strategy to anchor enzymes at injection sites via fusion to galectin-3 (G3), a carbohydrate-binding protein. Fusing G3 to luciferase extended bioluminescence in subcutaneous tissue to ~7 days, whereas unmodified luciferase was undetectable within hours. Engineering G3-luciferase fusions to self-assemble into a trimeric architecture extended bioluminescence in subcutaneous tissue to 14 days, and intramuscularly to 3 days. The longer local half-life of the trimeric assembly was likely due to its higher carbohydrate-binding affinity compared to the monomeric fusion. G3 fusions and trimeric assemblies lacked extracellular signaling activity of wild-type G3 and did not accumulate in blood after subcutaneous injection, suggesting low potential for deleterious off-site effects. G3-mediated anchoring to common tissue glycans is expected to be broadly applicable for improving local pharmacokinetics of various existing and emerging enzyme drugs.

[1] J. Crayton Pruitt Family Department of Biomedical Engineering, University of Florida, Gainesville, FL 32611, USA. Correspondence and requests for materials should be addressed to G.A.H. (email: ghudalla@bme.ufl.edu)

Presently, ~15% of all Food and Drug Administration (FDA)-approved proteins are enzymes used to treat various diseases, including lysosomal storage disorders[1], immuno-deficiency[2], leukemia[3], hemophilia B[4], and thrombosis[5]. Despite these notable successes, however, many attractive enzyme drug candidates fail in clinical trials due to unfavorable pharmacokinetics, pharmacodynamics, and safety profiles. For example, agalsidase alfa demonstrates widely varying pharmacokinetics[6], which requires frequent dosing that can lead to anti-drug antibody production[7]. A tumor-targeting variant of carboxypeptidase G2, a chemoprotective agent and chemotherapeutic pro-drug activator, has not been approved due to its immunogenicity[8]. Likewise, factor IX replacement therapy for hemophilia B is hindered by anti-drug antibodies that increase with the extent of mutation of the patient's factor IX gene[9]. Thus strategies to improve enzyme pharmacokinetics, pharmacodynamics, and safety profiles by extending half-life, increasing target site accumulation, and minimizing immunogenicity hold promise for increasing the number of FDA-approved enzyme drugs.

Various chemical modifications can extend enzyme half-life or increase accumulation within target tissues. For example, modifying enzymes with hydrophilic polymers (e.g., poly(ethylene glycol, PEG) or dextran) can extend half-life in circulation by increasing drug hydrodynamic radius to prevent renal clearance and by masking proteolytic degradation sites[10,11], as seen for Pegadamase and Pegaspargase[12]. However, modification with hydrophilic polymers does not promote enzyme accumulation at target sites within solid tissues and therefore is largely limited to enzymes that are effective in systemic circulation. Additionally, PEG and dextran conjugates may be immunogenic and can dramatically reduce enzyme drug activity[13,14]. Encapsulating enzymes within controlled-release vehicles or immobilizing them onto solid-phase carriers that can be introduced into a target tissue can extend the duration of localized biocatalysis, although the fabrication processes and degradation products of carriers and vehicles often greatly diminish enzyme catalytic activity[15]. Linking enzymes to antibodies or fragments thereof can increase accumulation within target tissues[16]. However, successful antibody-mediated targeting requires an antigen that is exclusively expressed by the target tissue, such as tumor antigens[17], as well as effective transport of relatively large antibody–enzyme conjugates over endothelial barriers, which is facilitated by the enhanced permeability and retention effect of tumor vasculature. Additionally, recombinant fusions of enzymes and single-chain antibodies are limited by weak binding affinity due to the lack of multivalent avidity effects, poor production efficiency, and potential immunogenicity[16,17]. In contrast, covalent conjugation of enzymes to whole antibodies affords limited control of drug stoichiometry and orientation, which together can diminish activity[18,19]. Finally, modifying enzymes with carbohydrates that recognize specific cell surface receptors has proven effective for increasing drug accumulation within target cell populations, as exemplified with Cerezyme used to treat Gaucher disease[20]. However, carbohydrate-mediated targeting can also lead to accumulation within non-target tissues, such as the liver and spleen, as seen for α-galactosidase A used to treat Fabry disease[6]. New strategies to enhance enzyme retention within tissues that address the practical limitations of targeted-mediated or vehicle-mediated delivery would afford significant opportunities to improve the therapeutic efficacy of many existing and emerging enzyme drugs.

Previous reports demonstrate that engineering growth factors or antibodies to bind extracellular matrix (ECM) proteins can increase their local retention in vitro and in vivo[21–24]. Herein we describe an alternative approach to prolong enzyme retention within tissues via recombinant fusion to human galectin-3

(G3) (Fig. 1a). G3 is a protein that binds to β-galactoside glycans, such as N-acetyllactosamine (LacNAc), as well as several glycosaminoglycans (GAGs), which are abundant within the extracellular space of mammalian tissues[25,26]. Extracellular G3 can modulate cell adhesion, migration, proliferation, differentiation, and death during various healthy and pathological processes by crosslinking cell surface glycoproteins into lattices[27]. Here we proposed to use G3 as a fusion domain to endow enzymes with affinity for extracellular carbohydrates to restrict their diffusion through the extracellular space (Fig. 1b–c), analogous to growth factor binding to ECM GAGs and proteins[28,29]. Thus we expect that enzyme–G3 fusions anchored to an injection site will be retained locally for a longer duration than freely diffusible enzymes, thereby leading to extended pharmacokinetics. Fusion to the N-terminus was chosen because the carbohydrate-recognition domain (CRD) of G3 is encoded by the C-terminal portion of the protein, while the N-terminal domain (NTD) is an

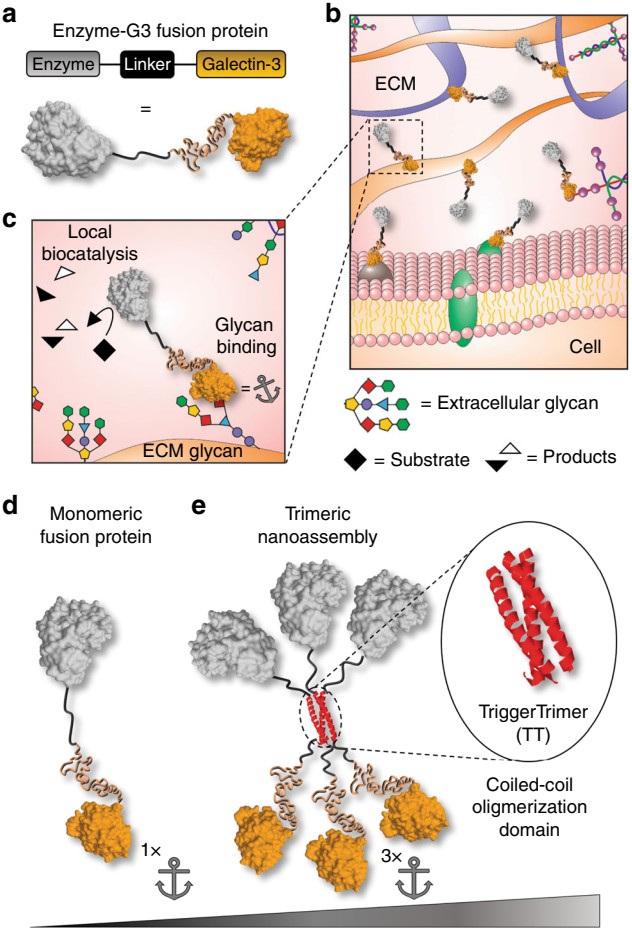

**Fig. 1** Design of G3 fusion proteins to locally anchor enzymes to tissues via extracellular glycan binding. **a–c** Schematic of recombinant enzymes fused with galectin-3 (i.e., enzyme-G3 fusion protein), which are anchored to an injection site via binding to cell surface and extracellular matrix (ECM) glycans. **d** Monomeric fusion protein consisting of an enzyme linked to the N-terminal domain of galectin-3 via a flexible peptide linker. **e** Trimeric nanoassembly formed by inserting the TT domain between the enzyme and G3 domains. The trimeric nanoassembly has higher glycan-binding affinity than the monomeric fusion protein due to multivalent avidity effects. PDB ID: 253L [10.2210/pdb253L/pdb] (generic enzyme), 2O7H [10.2210/pdb2O7H/pdb] (generic coiled-coil), and 5NF7 [10.2210/pdb5NF7/pdb] (carbohydrate-recognition domain of galectin-3)

intrinsically disordered domain thought to be involved in G3 self-association into higher-ordered oligomers[30]. Notably, G3 variants lacking the NTD retain carbohydrate-binding affinity yet lack wild-type (WT-G3) activity as an extracellular signal[31,32]. Thus we envisioned that fusing an enzyme to the G3 NTD would endow carbohydrate-binding affinity yet may also inactivate or alter native biological activities of G3 by disrupting its self-association into oligomers. G3 is an ideal fusion partner because it lacks disulfide bridges, does not require posttranslational modifications, and is relatively small. Finally, the extracellular carbohydrates recognized by G3 are highly conserved across mammalian species, suggesting that this anchoring strategy will be amenable for human and animal use without requiring significant redesign.

To characterize the carbohydrate-binding properties, bioactivity, and in vivo pharmacokinetics of G3 fusions, we created different constructs in which proteins, such as the bioluminescent reporter NanoLuc™ luciferase (NL)[33], were linked to the N-terminus of G3. The first, which we define as a monomeric fusion protein, consisted of a protein connected to G3 via a flexible linker peptide (Fig. 1d). The second consisted of a protein connected to G3 via a linker peptide that forms a three-stranded α-helical coiled-coil, referred to as TriggerTrimer (TT), which was previously developed via site-specific mutation of the GCN4 leucine zipper (i.e., GCN4-pM3)[34]. Protein-TT-G3 fusions were designed to self-assemble into a nano-scale structure having three protein and three G3 domains via the TT domain, which we define as a trimeric nanoassembly (Fig. 1e). The increased G3 avidity of protein-TT-G3 nanoassemblies confers higher carbohydrate-binding affinity, which anchors the enzyme to an injection site for a longer duration than monomeric G3 fusions.

## Results

**Expression of G3 fusion proteins**. To characterize the expression, assembly, and activity of G3 fusions, we created different constructs in which NL (~19 kDa), superfolder green fluorescent protein (GFP) (~27 kDa), or chondroitinase ABC I (ChABC) (~115 kDa) (i.e., small, medium, and large proteins, respectively) were linked to the N-terminus of G3 or TT (Fig. 2a). All fusion proteins were expressed and recovered from *Escherichia coli* in the soluble fraction at up to mg/L yields. Electrophoretic mobilities of these proteins under denaturing conditions were consistent with their theoretical denatured molecular weights (MWs), which ranged from 47.4 kDa for NL-G3 to 146.2 kDa for ChABC-TT-G3 (Supplementary Figure 1). The hydrodynamic size of each trimeric nanoassembly determined under native conditions via size-exclusion chromatography (SEC) was larger than its respective monomeric fusion, as indicated by a smaller elution volume (Fig. 2b). The empirical native MWs of each trimeric nanoassembly and fusion protein, which were determined from their elution volume, were consistent with native theoretical MWs (Supplementary Table 1). Hydrodynamic diameters, as determined via dynamic light scattering (DLS) number-weighted size distribution, were larger for each nanoassembly when compared to its respective monomeric fusion protein and increased as the MW of the enzyme increased (i.e., NL-G3 < GFP-G3 < ChABC-G3) (Fig. 2c). The DLS distribution demonstrated that monomeric fusion proteins and nanoassemblies were predominantly between 0 and 20 nm in diameter, suggesting that their tendency for non-specific aggregation in solution was low (Supplementary Figures 33–52). Consistent with this, protein concentration was similar before and after filtration through a 0.2-micron syringe filter (Supplementary Figures 16 and 27–32). Protein activities were evaluated for both monomeric fusion proteins and trimeric nanoassemblies when the concentration of the enzyme or GFP

domain was held constant (Fig. 2d). NL-G3 and NL-TT-G3 produced comparable bioluminescence to equimolar quantities of wild-type NL (WT-NL) in the presence of the NL substrate furimazine (Supplementary Figure 2), similar to previously reported soluble NL fusion proteins[35]. Likewise, GFP-G3 and GFP-TT-G3 produced comparable fluorescence when the GFP concentration was held constant, while ChABC-G3 and ChABC-TT-G3 demonstrated comparable catalytic activity for degrading chondroitin sulfate (CS)-A (Fig. 2d). Collectively, these data demonstrated that active proteins with a broad range of MWs can be expressed as G3 fusions in microbial hosts and assembled into trimeric nanoassemblies via the TT domain.

**Carbohydrate-binding properties of G3 fusion proteins**. We then compared the lactose binding properties of wild-type G3 (WT-G3), monomeric G3 fusion proteins, and trimeric G3 nanoassemblies using lactose affinity chromatography (Supplementary Figure 3). WT-G3 and monomeric G3 fusion proteins eluted from the column at a comparable concentration of soluble lactose indicating that they had similar binding affinity for immobilized lactose. In contrast, trimeric G3 nanoassemblies eluted with much broader profiles shifted to higher soluble lactose concentrations, corresponding with a higher apparent binding affinity for immobilized lactose than WT-G3 and monomeric fusions.

Next, we used NL luminescence and GFP fluorescence to characterize binding of monomeric fusion proteins and trimeric nanoassemblies to different surface-adsorbed ECM glycoproteins and proteoglycans, collectively referred to as glycoconjugates. We first compared fusion protein binding to surface-adsorbed asialofetuin (ASF), laminin, and collagen IV that are decorated with β-galactosides[25,36], aggrecan that is decorated with G3-binding CS GAGs[26], and collagen I that is minimally glycosylated and not an observed G3 ligand[25,37]. More trimeric nanoassemblies bound to surfaces coated with ASF, laminin, collagen IV, and aggrecan than control surfaces lacking adsorbed glycoconjugates, and this binding was inhibited by soluble LacNAc suggesting that it was mediated by specific interactions between glycans on adsorbed glycoconjugates and the CRD of G3 (Fig. 3a–c). Likewise, no binding of NL-TT-G3 or GFP-TT-G3 to collagen I was observed, further suggesting that binding was specifically mediated by interactions between glycoconjugates and the CRD of G3 (Fig. 3a, b). No significant binding was detected for monomeric fusion proteins incubated with adsorbed glycoconjugates at an equivalent concentration of NL or GFP as the trimeric nanoassemblies (Fig. 3a, b). However, NL-G3 and GFP-G3 could be visualized bound to laminin at higher concentrations and this binding was also inhibited by soluble LacNAc suggesting that it was mediated by specific interactions between laminin glycans and the G3 CRD (Fig. 3c). Collectively, these data demonstrate that G3 fusion proteins and nanoassemblies specifically recognize glycoconjugates decorated with G3-binding glycans and that nanoassemblies may have higher binding affinity than monomeric fusions, as reflected by their greater extent of binding.

To further characterize the binding affinity of monomeric fusion proteins and trimeric nanoassemblies for different glycoconjugates, we evaluated competitive inhibition of their binding to ASF by soluble LacNAc, as well as their saturation binding profile for surface-adsorbed ASF and laminin. Higher concentrations of LacNAc were required to inhibit binding of NL-TT-G3 to adsorbed ASF when compared to NL-G3 (Fig. 3d). Likewise, more GFP-TT-G3 bound to ASF or laminin than GFP-G3 over a range of GFP concentrations, with GFP-TT-G3 approaching saturation at ~2 μM while GFP-G3 did not reach

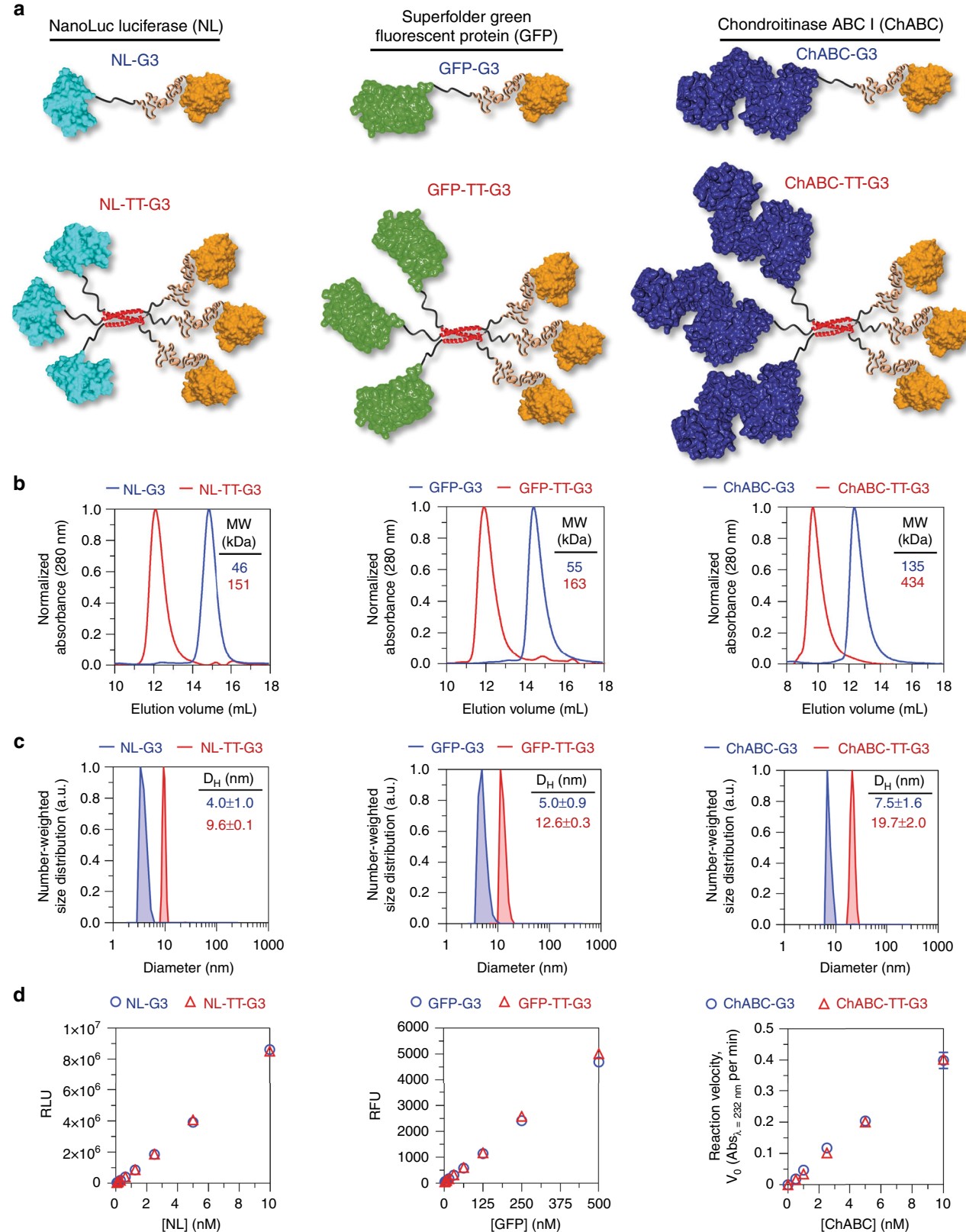

**Fig. 2** Design and characterization of monomeric G3 fusion proteins and trimeric nanoassemblies. **a** Predicted structure of NanoLuc[TM] luciferase (NL), superfolder green fluorescent protein (GFP), and chondroitinase ABC I (ChABC) monomeric G3 fusion proteins and trimeric nanoassemblies. **b** Approximate molecular weight determined under native conditions via size-exclusion chromatography. **c** Average hydrodynamic diameter estimated via dynamic light scattering. **d** Quantitative bioluminescence, fluorescence, and reaction velocity of NL, GFP, and ChABC fusions, respectively. PDB ID: 5IBO [10.2210/pdb5IBO/pdb] (NanoLuc[TM] luciferase), 2B3P [10.2210/pdb2B3P/pdb] (superfolder GFP), 1HN0 [10.2210/pdb1HN0/pdb] (ChABC) for **a**. $N \geq 3$, mean ± s.d. for **c**. $N = 3$, mean ± s.d. for **d**. Data for monomeric G3 fusion proteins appear as blue circles/traces, trimeric nanoassemblies as red triangles/traces

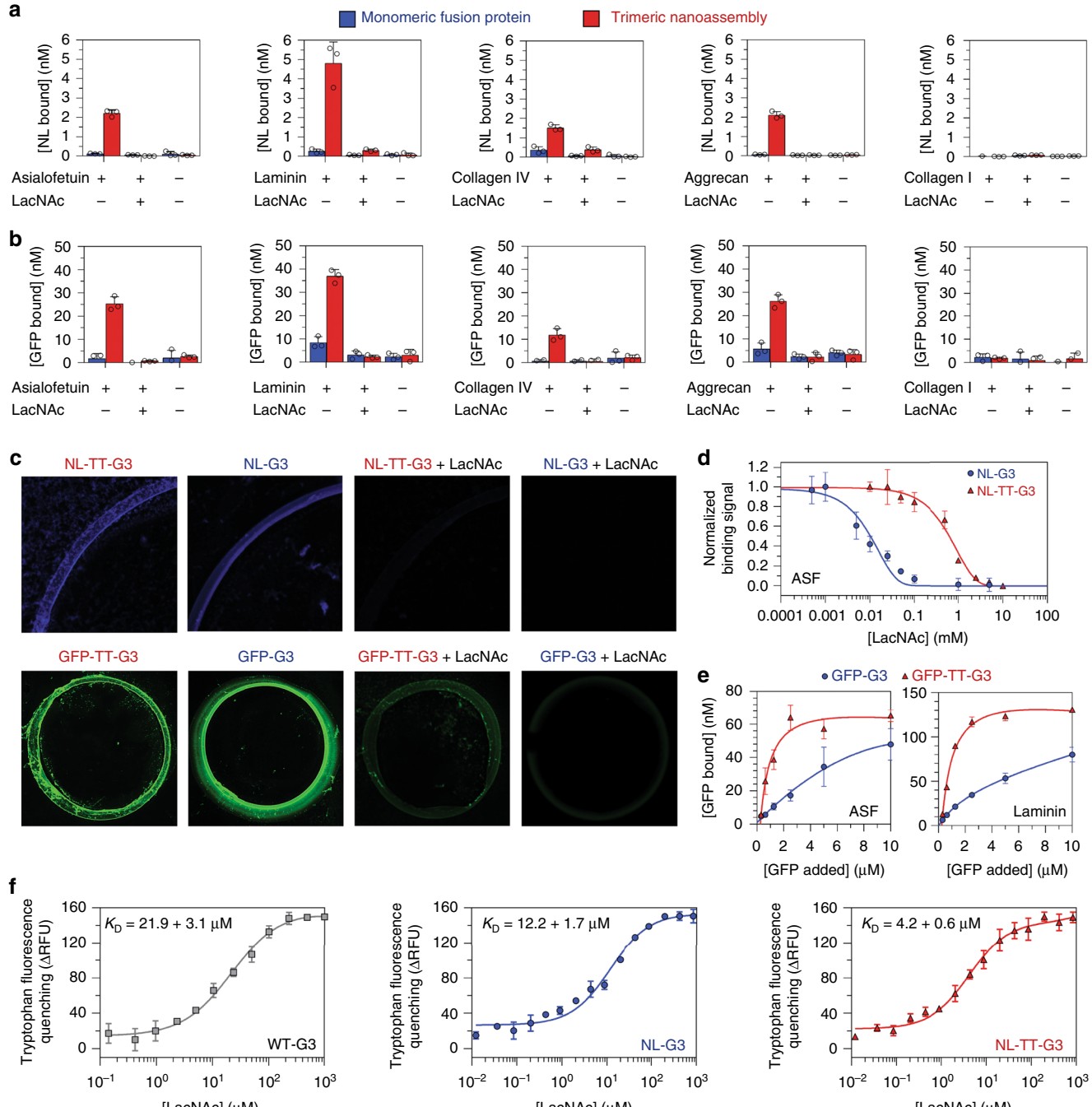

**Fig. 3** Carbohydrate-binding properties of monomeric G3 fusion proteins and trimeric nanoassemblies. **a**, **b** NL-G3, NL-TT-G3, GFP-G3, and GFP-TT-G3 binding to asialofetuin (ASF), laminin, collagen IV, aggrecan, and collagen I (negative control) adsorbed onto plastic. **c** Micrographs taken of NL luminescence or GFP fluorescence localized to a laminin coffee ring adsorbed onto glass. Fusion proteins were mixed with soluble LacNAc to demonstrate inhibition of binding to adsorbed glycoconjugates for **a**–**c**. **d** Competitive inhibition of binding of NL-G3 and NL-TT-G3 to adsorbed ASF by soluble LacNAc. **e** Saturation binding of GFP-G3 and GFP-TT-G3 to adsorbed ASF and laminin. **f** Tryptophan fluorescence quenching of wild-type G3 (WT-G3), NL-G3, and NL-TT-G3 via binding to soluble LacNAc. $N = 3$, mean ± s.d. for **a**, **b**, **d**–**f**. Data points at or above baseline signal are shown as open circles in **a**, **b**. Data for WT-G3 appear as gray squares/traces, for monomeric G3 fusion proteins as blue bars/circles/traces, and trimeric nanoassemblies as red bars/triangles/traces

saturation up to 10 μM (Fig. 3e). Scatchard analysis of these saturating binding data suggested that GFP-G3 interactions with adsorbed glycoconjugates were non-cooperative, which was expected because the monomeric fusion protein only has one CRD (Supplementary Figure 4). The (1:1) dissociation constant ($K_D$) of GFP-G3 for ASF and laminin was predicted to be 11.2 and 5.6 μM, respectively, where these differences likely reflect

differences in the number of glycans conjugated to ASF versus laminin or the amount of each glycoprotein adsorbed onto the surface. In contrast, Scatchard analysis suggested positive cooperativity for GFP-TT-G3 interactions with ASF and laminin glycans (Supplementary Figure 4), indicating that two or three CRDs of the trimeric nanoassembly may be bound simultaneously, which precluded accurate estimation of NL-TT-G3 $K_D$

for ASF and laminin. Taken together with the apparent binding affinity of monomeric fusions and trimeric nanoassemblies for immobilized lactose (Supplementary Figure 3), these observations supported our overall hypothesis that multivalent avidity effects can endow trimeric nanoassemblies with higher apparent carbohydrate-binding affinity than monovalent fusion proteins.

Next, we compared binding of NL-G3 and NL-TT-G3 to various sulfated GAGs that are known to bind WT-G3[26]. Specifically, we used a competition assay in which NL-G3 or NL-TT-G3 was first mixed with CS-A, CS-B, CS-C, or heparin and then added to laminin-coated plates. All GAGs competitively inhibited NL-TT-G3 and NL-G3 binding to laminin, albeit to different extents, and the percentage of NL-TT-G3 bound to GAG versus laminin was greater than that for NL-G3 in all cases (Supplementary Figure 5). We expected that NL-G3 and NL-TT-G3 would be resistant to non-specific interactions with anionic GAGs due to their net charges of −6 and -5 at neutral pH, respectively. Likewise, as shown in Fig. 3a, b, soluble LacNAc inhibited NL-TT-G3 and GFP-TT-G3 binding to aggrecan, suggesting that trimeric nanoassemblies specifically recognize CS GAGs. Thus, taken together, these data demonstrated that NL-G3 and NL-TT-G3 recognize various GAGs that are known to bind WT-G3, although with different apparent affinities likely due to differences in their carbohydrate composition or sulfation profile. These data also suggested a propensity for NL-TT-G3 to remain bound to the first ligand it encountered. Importantly, these results further supported our hypothesis that avidity effects can endow G3 nanoassemblies with higher relative binding affinity for carbohydrates than monomeric fusion proteins.

Finally, to determine whether the observed increase in NL-TT-G3 carbohydrate-binding affinity was due solely to avidity effects or was due in part to perturbation of G3 conformation, we characterized binding of soluble, monovalent lactose or LacNAc to NL-TT-G3, NL-G3, and WT-G3 using tryptophan fluorescence quenching. Unexpectedly, the LacNAc:NL-TT-G3 $K_D$ and the LacNAc:NL-G3 $K_D$ were approximately five- and two-fold lower than the LacNAc:G3 $K_D$, respectively (Fig. 3f). Furthermore, the lactose:NL-TT-G3 $K_D$ was significantly lower than that of both NL-G3 and WT-G3, which were not statistically different from each other (Supplementary Figure 6), and were consistent with previous reports for G3 and G3 fusions[25,38]. This may be due to regulation of the G3 CRD by its NTD. Some studies suggest that the NTD can mask the CRD[39,40], while others suggest that transient intramolecular interactions between the NTD and CRD of WT-G3 can diminish carbohydrate binding[41]. Consistent with this, some G3 fragments with truncated NTDs demonstrated significantly higher affinity for carbohydrates than WT-G3[42,43]. We postulate that NTD fusion to NL or TT may hinder NTD–CRD interactions, thereby increasing G3 affinity for LacNAc. Additionally, steric hindrance imposed by NL-TT-G3 oligomerization may further prevent NTD–CRD interactions, resulting in NL-TT-G3 having higher LacNAc and lactose-binding affinity. Although the actual molecular mechanism underlying this phenomenon remains unknown, our observations suggest that increased apparent affinity of NL-TT-G3 for extracellular glycoconjugates is likely due to both multivalent avidity effects and increased monovalent carbohydrate-binding affinity.

**Extracellular signaling activity of G3 fusion proteins**. Extracellular WT-G3 can induce apoptosis of T cells, indicated by phosphatidylserine exposure, loss of metabolic activity, permeability to propidium iodide (PI), and DNA fragmentation[44], which could be a deleterious immunosuppressive side effect of enzyme–G3 fusion proteins. We used a combination of assays to compare changes in Jurkat T cell behavior induced by WT-G3, NL-G3, and NL-TT-G3. Jurkat T cells treated with NL-G3 and NL-TT-G3 produced luminescence in the presence of furimazine (Fig. 4a), while cells treated with GFP-G3 and GFP-TT-G3 produced fluorescence (Fig. 4b), demonstrating that both fusion proteins and nanoassemblies bound to cell surface glycans. Notably, significantly more trimeric nanoassemblies bound to Jurkat T cells than monomeric fusion proteins (Fig. 4c), consistent with glycoconjugate-binding data (Fig. 3a, b, d, e). As expected, WT-G3 induced Jurkat T cell agglutination (Fig. 4d), exposure of phosphatidylserine, and permeability to PI (Supplementary Figures 7-8), as well as metabolic activity loss (Fig. 4e), collectively indicating that WT-G3 decreased Jurkat T cell viability. In contrast, neither NL-G3 nor NL-TT-G3 induced Jurkat T cell agglutination nor metabolic activity loss at an equimolar G3 dose (Fig. 4d, e). Interestingly, Jurkat T cells treated with NL-G3 were negative for phosphatidylserine exposure and demonstrated comparable PI permeability to untreated cells, whereas cells treated with NL-TT-G3 were positive for phosphatidylserine exposure, yet demonstrated comparable PI permeability to untreated cells (Supplementary Figures 7-8). Although often taken as an early marker of apoptosis, WT-G3 has been shown to induce non-apoptotic phosphatidylserine exposure on MOLT-4 leukemic T cells[44], characterized by Annexin V positive/PI negative cells, and phosphatidylserine exposure independent of apoptosis has been reported for activated CD8+ T cells[45]. The comparable metabolic activity of Jurkat T cells treated with NL-TT-G3 and untreated cells suggests that phosphatidylserine exposure may not be an accurate determinant of early apoptosis in this model. Thus these data demonstrate that G3 fusions and nanoassemblies do not share WT-G3 activity for inducing Jurkat T cell agglutination, membrane permeability, and loss of metabolic activity, although NL-TT-G3 can induce phosphatidylserine exposure.

In addition to inducing apoptosis, lower concentrations of WT-G3 can also induce Jurkat T cell secretion of interleukin (IL)-2[46], a cytokine that can promote differentiation of effector and memory T cell populations upon antigen recognition. Jurkat T cells treated with WT-G3 secreted significantly more IL-2 than untreated cells (phosphate-buffered saline (PBS)) or cells treated with WT-G3 plus lactose inhibitor (Fig. 4f). In contrast, Jurkat T cells treated with NL-G3 and NL-TT-G3 secreted comparable amounts of IL-2 as untreated cells or cells treated with NL-G3 or NL-TT-G3 plus lactose inhibitor. Thus NL-G3 and NL-TT-G3 lacked the activity of WT-G3 for inducing Jurkat T cell IL-2 secretion.

The observation that NL-G3 and NL-TT-G3 bound to Jurkat T cells, yet had diminished activity for inducing cell agglutination, loss of metabolic activity, and IL-2 secretion when compared to WT-G3 may be due to the fused cargo on the NTD of G3. WT-G3 can self-associate into oligomers upon binding glycoproteins via interactions involving its NTD[47,48], and one proposed mechanism of G3 activation of outside–in signaling is through cell surface glycoprotein clustering via these oligomers[49]. Prior reports demonstrated that G3 mutants with truncated NTDs failed to induce T cell death, likely because they were unable to self-associate into higher-ordered oligomers[31,32]. Here we considered that NL-G3 and NL-TT-G3 may also fail to induce Jurkat T cell agglutination, metabolic activity loss, and cytokine secretion because of a diminished ability to cluster glycoproteins when compared to WT-G3. Regions of punctate staining, suggestive of surface glycan crosslinking, were identified in fluorescent photomicrographs of GFP-TT-G3 bound to Jurkat T cells, yet were less pronounced in micrographs of bound GFP-G3 (Fig. 4b). Because G3 crosslinking of cell surface glycans is difficult to evaluate quantitatively, we further characterized glycan

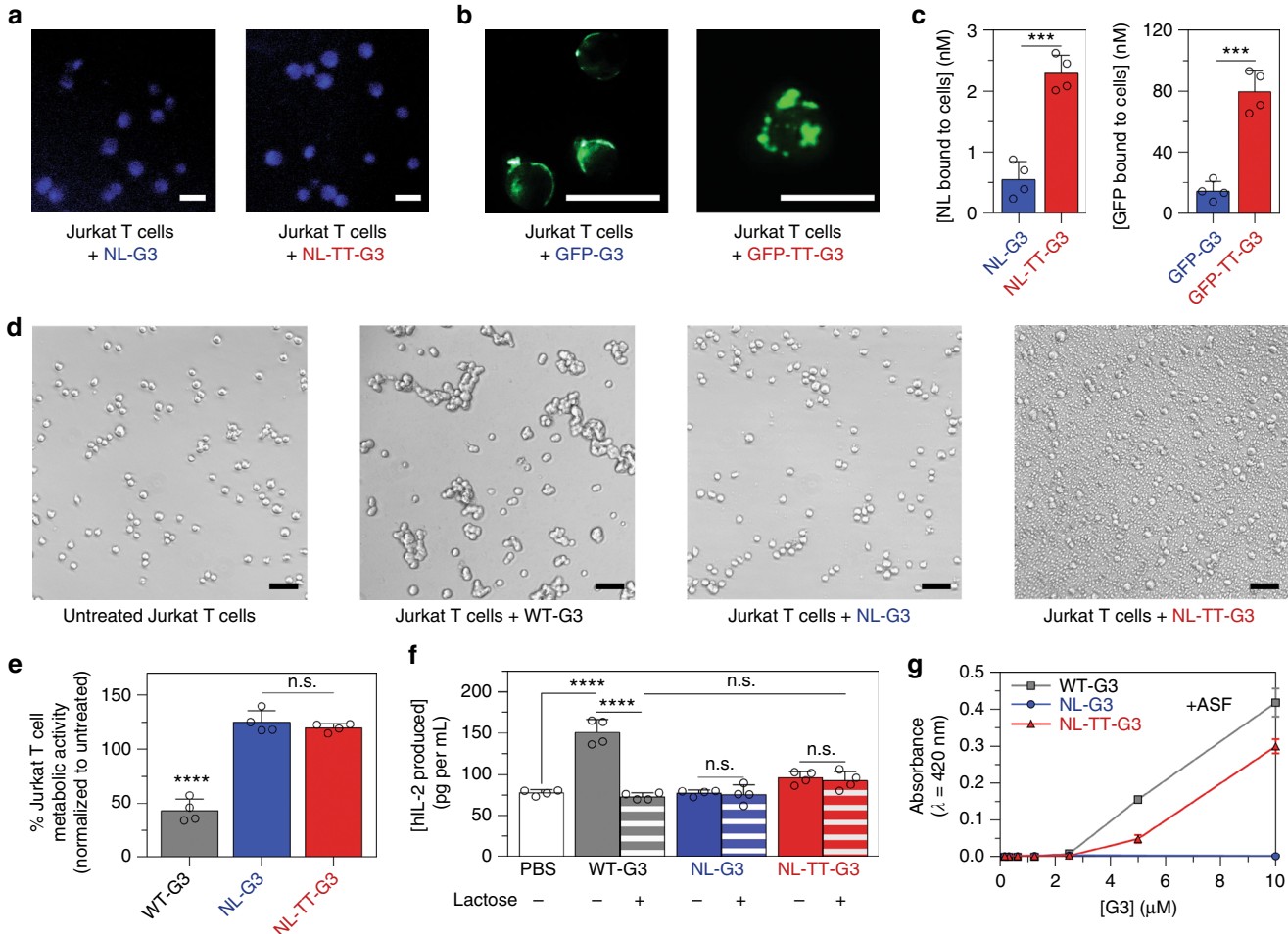

**Fig. 4** Extracellular signaling activity of monomeric G3 fusion proteins and trimeric nanoassemblies. **a** Micrographs demonstrating NL-G3 and NL-TT-G3 bioluminescence on Jurkat T cells using a blue fluorescence filter. **b** Micrographs demonstrating GFP-G3 and GFP-TT-G3 fluorescence on Jurkat T cells using a green fluorescence filter. **c** Amount of NL or GFP bound to Jurkat T cells after 4 h incubation with NL-G3, NL-TT-G3, GFP-G3, or GFP-TT-G3. **d** Bright-field micrographs of Jurkat T cells incubated with PBS (untreated, negative control), WT-G3 (positive control), NL-G3, or NL-TT-G3 for 4 h. **e** Percentage of metabolic activity of Jurkat T cells incubated with WT-G3 (positive control), NL-G3, or NL-TT-G3 for 4 h. **f** hIL-2 produced by Jurkat T cells after incubation with PBS (negative control), WT-G3 (positive control), NL-G3, or NL-TT-G3 for 24 h. Lactose was added as an inhibitor of G3 binding to Jurkat T cells. **g** Asialofetuin (ASF) (7 μM) precipitation with various quantities of WT-G3, NL-G3, or NL-TT-G3 as measured by light scattering and absorbance at 420 nm. $N = 4$, mean ± s.d., ***$p < 0.001$, Student's $t$ test for **c**. $N = 4$, mean ± s.d., n.s. is no significant differences, ****$p < 0.0001$, ANOVA with Tukey's post hoc for **e** and **f**. $N = 3$, mean ± s.d. for **g**. Scale bar = 25 μm for **a**, **b**. Scale bar = 50 μm for **d**. Data points at or above baseline signal are shown as open circles in **c**, **e**, **f**. Data for WT-G3 appear as gray bars/squares/traces, for monomeric G3 fusion proteins as blue bars/circles/traces, and for trimeric nanoassemblies as red bars/triangles/traces

crosslinking via WT-G3, NL-G3, and NL-TT-G3 by adapting established precipitation assays based on the model glycoprotein ASF[47,50]. WT-G3 crosslinked ASF into insoluble precipitates (Fig. 4g and Supplementary Figures 53-55 and 62-64), consistent with prior reports[50,51]. In contrast, NL-G3 failed to crosslink ASF into insoluble precipitates at any concentration tested (Fig. 4g and Supplementary Figures 56-58 and 65-67). Interestingly, NL-TT-G3 crosslinked ASF into insoluble precipitates (Fig. 4g and Supplementary Figures 59-61 and 68-70), but the resulting aggregates were smaller and took longer to form than those produced by WT-G3 (Supplementary Figures 9-11). Additionally, NL-TT-G3 and GFP-TT-G3 induced formation of micron-sized particles in the presence of 10% fetal bovine serum (Fig. 4d and Supplementary Figure 12), likely by interacting with G3-binding serum glycoproteins such as α-2-macroglobulin[52], whereas WT-G3 at an equivalent concentration induced less serum glycoprotein crosslinking (Fig. 4d and Supplementary Figure 12). Collectively, these data suggest that appending a protein onto the G3 NTD can alter its glycoprotein crosslinking properties, presumably by perturbing its self-association into higher-ordered oligomers. Engineering G3 self-association into a trimer via the TT domain restored some of its glycoprotein crosslinking properties, yet this construct did not induce Jurkat T cell agglutination, PI permeability, loss of metabolic activity, or IL-2 secretion. Thus WT-G3 activity as an extracellular T cell signal may require a CRD valency >3 or a different CRD organization than that afforded by oligomerization via the TT domain. Alternatively, partitioning of NL-TT-G3 or GFP-TT-G3 into assemblies with serum glycoproteins may decrease the amount of G3 bound to cell surface glycans below the threshold required for activation of signaling mechanisms that lead to agglutination, loss of metabolic activity, PI permeability, or IL-2 secretion. Taken together, these data suggest that NL-G3 and NL-TT-G3 will have a low likelihood of inducing unwanted changes in T cell behavior in vivo, in part due to an altered ability to crosslink and cluster cell surface glycoproteins.

**Local in vivo half-life of G3 fusion proteins**. We evaluated WT-NL, NL-G3, and NL-TT-G3 pharmacokinetics in vivo at different common injection sites in mice by measuring local bioluminescence over time via in vivo imaging. Mice received a single, equivalent enzyme molar dose of WT-NL, NL-G3, or NL-TT-G3 subcutaneously into the hock and scruff, as well as intramuscularly (IM) into the caudal thigh muscle, followed by daily local injections of furimazine substrate. Localized bioluminescence was detectable at subcutaneous sites in mice that received NL-TT-G3 for approximately 14 days, whereas catalytic activity persisted for approximately 6–8 days in mice that received NL-G3 (Fig. 5a, b). In contrast, no bioluminescence was detectable at 24 h in mice that received WT-NL. Localized bioluminescence was detectable at an IM site for approximately 3 days in mice that received NL-TT-G3, while catalytic activity persisted for approximately 1–2 days in mice that received NL-G3 (Fig. 5a, b). Again, no bioluminescence was detectable at an IM site at 24 h in mice that received WT-NL. NL-TT-G3 half-life at subcutaneous sites (mean ± standard deviation, 33.5 ± 13.3 h, hock; 31.8 ± 24.4 h, scruff) was significantly longer than that of NL-G3 (mean ±

standard deviation, 10.4 ± 2.8 h, hock; 6.1 ± 1.0 h, scruff), whereas half-life could not be accurately determined for WT-NL due to rapid signal loss within 24 h (Fig. 5c). Likewise, NL-TT-G3 half-life at an IM site was 4.2 ± 1.8 h (mean ± standard deviation) (Fig. 5c), while NL-G3 and WT-NL half-life could not be accurately determined. Taken together, these data demonstrated that fusion to G3 can anchor an enzyme at an injection site for a significantly longer duration than unmodified enzyme and that nanoassemblies have a longer half-life than monomeric fusion proteins.

Given that NL-G3 and NL-TT-G3 hydrodynamic diameters were smaller than typical ECM pore diameters (tens of nm–μm)[53], we assumed that prolonged nanoassembly retention was not due to its larger size. Rather, we inferred from these data that the longer half-life of NL-TT-G3 relative to NL-G3 was primarily due to differences in their apparent carbohydrate-binding affinity, as observed in vitro (Figs. 3, 4c). The differences in NL-TT-G3 half-life at subcutaneous and IM injections sites may be due to attenuation of blue light emitted by NL at the deeper IM injection site[54]. The significant differences in NL-TT-G3 half-life at

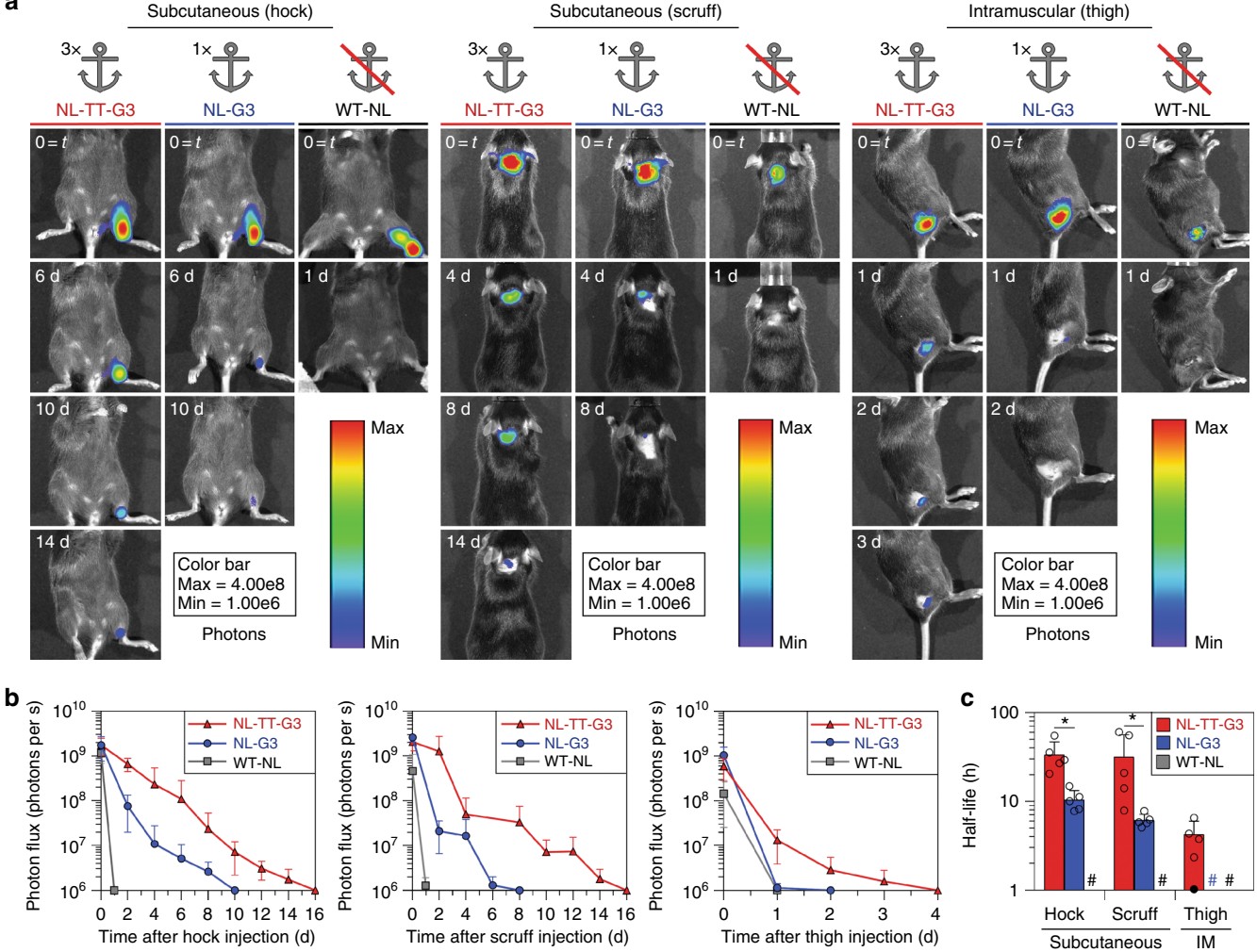

**Fig. 5** Injection site half-life of monomeric G3 fusion proteins and trimeric nanoassemblies. **a** Bioluminescence images and **b** photon flux at various time points for mice that received NL-TT-G3, NL-G3, or WT-NL (equivalent moles of NL) in the hock, scruff, or thigh. **c** Biocatalytic activity half-life of NL-TT-G3, NL-G3, or WT-NL in different tissues. $N = 5$, mean ± s.d., *$p < 0.05$, ANOVA with Tukey's post hoc. In **c**, open circles represent each half-life calculation, closed circle represents individual animals for which half-life could not be accurately calculated due to rapid enzyme clearance from the injection site, and hash sign represents groups for which half-life could not be accurately calculated due to rapid enzyme clearance from the injection site (signal at $t = 24$ was at baseline for ≥3 animals in a cohort). Data for WT-NL appear as gray bars/squares/traces, NL-G3 as blue bars/circles/traces, and NL-TT-G3 as red bars/triangles/traces

subcutaneous and IM sites may also reflect differences in glycan content of different tissues, which could therefore be a key determinant of G3 fusion protein and nanoassembly pharmacokinetics. Thus the ability to increase the apparent carbohydrate-binding affinity of enzyme-G3 fusion proteins via self-assembly into multivalent structures may provide a simple route to extend enzyme pharmacokinetics even within tissues having low glycan content.

**Clearance of G3 fusion proteins.** The efficacy of enzymes as drugs is frequently challenged by various clearance mechanisms including phagocytosis by reticuloendothelial cells, antibody neutralization, proteolytic degradation, or renal excretion. To identify possible mechanisms of NL-G3 and NL-TT-G3 clearance, we first evaluated whether NL was present in circulation at various time points after subcutaneous injection into the hock. Less than 0.03% of the total injected mass of NL-TT-G3 or NL-G3 was detected in blood at 6 h, whereas no luminescence was detected in blood at 24 h or daily thereafter (Fig. 6a). This suggested that the enzyme was not entering circulation, enzyme concentration in blood was too dilute to detect (<pM) as a result of rapid glomerular filtration, or that the enzyme was inactivated by serum proteases. However, NL-TT-G3 and NL-G3 were active for >18 h in 25% mouse serum at 37 °C in vitro, suggesting that degradation by serum proteases was likely not a clearance mechanism (Fig. 6b). Given that the hydrodynamic diameter of NL-TT-G3 exceeded that of glomerular pores (4.5–5 nm)[55], we inferred that the absence of NL in the blood was not due to renal filtration. Rather, these data suggested that NL-G3 and NL-TT-G3 were likely eliminated locally.

Matrix metalloproteinases (MMPs), such as collagenase, can cleave the collagen-like NTD of WT-G3, thereby dissociating it from the ~16 kDa CRD (Fig. 6c)[56]. Here we characterized collagenase degradation of WT-G3, NL-G3, and NL-TT-G3 using denaturing gel electrophoresis and SEC. As expected, collagenase-treated WT-G3 migrated as a single band having a lower MW of ~16 kDa (Fig. 6d), consistent with prior reports[36], and eluted at a higher volume fraction (i.e., lower MW) than untreated WT-G3 via SEC (Supplementary Figure 13). Similarly, collagenase-treated NL-G3 and NL-TT-G3 migrated to lower MWs when compared to untreated protein (Fig. 6d) and eluted at higher volume fractions via SEC (Supplementary Figure 13). Taken together, these data suggest that the decreasing localized biocatalysis observed over time in vivo may be due to MMP-mediated dissociation of NL from G3, which would lead to enzyme diffusion away from the injection site. Additionally, other proteolytic enzymes not assayed here could also degrade NL or the linker domain, which would also lead to decreased localized biocatalysis over time in vivo. Thus, collectively, these data suggest that enzyme–G3 fusion protein and nanoassembly half-life will depend, at least in part, on their susceptibility to degradation via local tissue proteases.

Finally, because these fusion proteins are foreign to the host, we evaluated generation of anti-NL antibodies raised by mice that received subcutaneous NL-G3 or NL-TT-G3 twice, 4 weeks apart. These time points were chosen to ensure that protein injected at the initial time point was completely cleared before the second dose was received and to allow for sufficient time for serum immunoglobulin G (IgG) antibody generation. C57BL/6 mice that received NL emulsified in TiterMax Gold™ adjuvant raised significant serum IgG reactive against NL, while mice that received a TT-GFP fusion lacking G3 emulsified in TiterMax Gold™ adjuvant also raised significant serum IgG that were likely reactive against both the TT and GFP domains (Supplementary Figure 14). In contrast, C57BL/6 mice raised little to no IgG

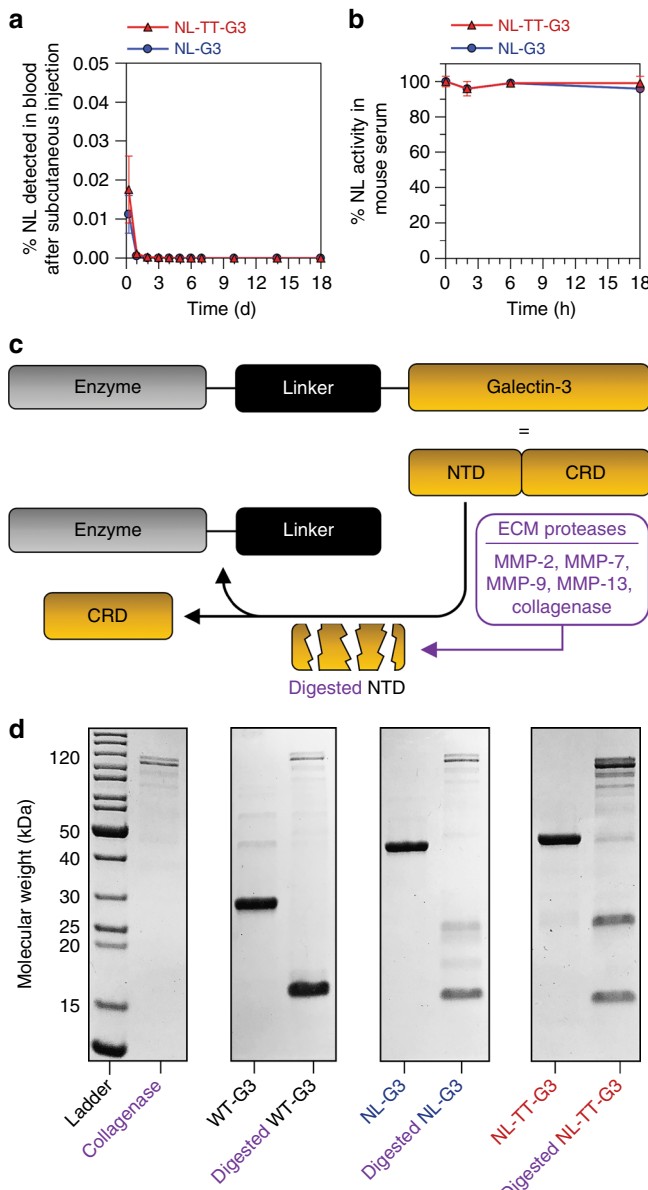

**Fig. 6** Circulating concentration and proteolysis of monomeric G3 fusion proteins and trimeric nanoassemblies. **a** Percentage of NL, by total mass injected, in blood samples collected over time after subcutaneous injection into the hock. **b** Percentage of NL activity, relative to NL activity at the initial time point, in 25% mouse serum in vitro. **c** Schematic of ECM proteases, such as collagenase and other MMPs, digesting the N-terminal domain (NTD) of G3, thereby separating the carbohydrate-recognition domain (CRD) of G3 from the enzyme fusion partner. **d** SDS-PAGE analysis of collagenase-mediated digestion of WT-G3, NL-G3, and NL-TT-G3. Uncropped gel image found in Supplementary Figure 26. $N = 5$, mean ± s.d. for **a**. $N = 3$, mean ± s.d. for **b**. Data for NL-G3 appear as blue circles/traces and NL-TT-G3 as red triangles/traces

serum antibodies against NL, NL-G3, or NL-TT-G3 when the proteins were administered in the absence of an immunostimulatory adjuvant (Supplementary Figure 14). The lack of antibodies reactive toward G3 fusions after two injections suggested that any depot effect due to the G3 domain did not enhance anti-NL immunogenicity and, therefore, was likely not a mechanism of clearance in these studies. However, future efforts will likely be needed to thoroughly evaluate the immunogenicity of any new G3 fusion proteins on a case-by-case basis, for example by

subjecting hosts to repeated injections at doses and over time frames that are relevant for their intended use.

## Discussion

In recent years, galectin fusions have gained interest as tools for glycobiology. For example, fusions of G3 with SNAP-tag and fluorescent proteins were developed for enzyme-linked immunosorbent assay (ELISA) and flow cytometric applications[38,57], while G3 fusions with bacterial alkaline phosphatase were developed to identify tissue glycosylation patterns[58]. Likewise, galectin-1 fusion domains are finding use for increasing the solubility of a glycosyltransferase enzyme that is prone to aggregation in *E. coli*, analogous to the application of recombinant maltose-binding fusion proteins[59]. Here we add to these applications by demonstrating that fusing an enzyme to G3 can provide prolonged localized biocatalysis proximal to minimally invasive injection sites by endowing the enzyme with affinity for extracellular glycans.

Prominent paradigms for therapeutic enzyme delivery extend half-life in circulation or rely on exit from circulation at specific tissue sites through discontinuous, damaged, or dysfunctional vasculature. The approach reported here establishes a delivery modality in which enzymes introduced directly into a desired tissue site persist over a tunable duration of time. Local enzyme activity half-life is dependent on the number of G3 units, which can be varied by engineering enzyme–G3 fusions to self-assemble into multimeric nanoassemblies. Importantly, WT-G3 activity for inducing T cell agglutination, phosphatidylserine exposure, loss of metabolic activity, IL-2 secretion, and death was abolished by fusing an enzyme to its N-terminus, thus potential immunomodulatory side effects were mitigated. Engineering G3 self-association into a trimeric architecture increased carbohydrate-binding affinity, yet was insufficient to restore G3 activity as an extracellular signal, suggesting that WT-G3 biological activity requires oligomers having a CRD valency >3 or a different CRD orientation than that afforded by assembly via the TT domain. Notably, small, medium, and large G3 fusion proteins and nanoassemblies were expressed and recovered in the soluble fraction from microbial hosts, suggesting that this platform is widely applicable to different functional protein domains and can be readily scaled up using established, cost-effective bioprocess methods. Additionally, in contrast to many conventional drug targeting moieties, G3 recognizes carbohydrate ligands that are highly conserved across mammalian species, suggesting that this anchoring strategy will be translatable across humans and animals. Finally, although not demonstrated here, fusion to G3 will likely be applicable for localizing activity of non-enzyme protein drugs within specific tissues. Thus, by providing an attractive general approach to locally anchor proteins within target tissues via minimally invasive injection routes while also preventing systemic biodistribution, we envision that galectin-3 fusions will be broadly useful for improving local pharmacokinetics of various emerging therapeutic enzymes as well as those that have stalled in the development pipeline.

## Methods

**Recombinant protein expression and purification.** NanoLuc™ is the tradename of an engineered deep sea shrimp luciferase variant developed by Promega Corporation[33]. Genes encoding fusion proteins were inserted into pET-21d(+) vectors between NcoI and XhoI sites. All genetic and protein sequences are provided in Supplementary Notes 1-2. Plasmids were first transformed into One Shot™ TOP10 Chemically Competent *E. coli* (ThermoFisher) and selected on ampicillin (100 μg/mL) doped LB/agar plates overnight at 37 °C. Isolated colonies from the plates were picked and cultured in 5 mL LB broth with ampicillin (100 μg/mL) overnight in an orbital shaker at 37 °C, 225 rpm. Recombinant DNA was recovered with a Plasmid Miniprep Kit (Qiagen) and sequenced using the Sanger method. Positive DNA sequences were then transformed into Origami™ B (DE3) *E. coli*

(Novagen) and selected on ampicillin (100 μg/mL) and kanamycin B (50 μg/mL) doped LB/agar plates overnight at 37 °C. Positive clones were picked and used to inoculate 5 mL of LB broth containing ampicillin (100 μg/mL) and kanamycin B (50 μg/mL). Cultures were grown overnight at 37 °C, 225 rpm on an orbital shaker and then sub-cultured into 1 L 2×TY media (16 g tryptone, 10 g yeast extract, 5 g NaCl) with ampicillin (100 μg/mL) and kanamycin B (50 μg/mL) at 37 °C, 225 rpm in an orbital shaker until an O.D. at 600 nm = 0.6–0.8 was reached. Cultures were then supplemented with 0.5 mM isopropyl β-D-1-thiogalactopyranoside (ThermoFisher) to induce protein expression and incubated for 18 h in an orbital shaker at 18 °C, 225 rpm. Bacteria were washed with PBS via centrifugation (11,300 × *g* at 4 °C for 10 min) with a Sorvall RC 6 Plus Superspeed Centrifuge (ThermoFisher). Afterwards, the cell pellet was incubated for 20 min at room temperature (RT) with lysis buffer: B-PER™ bacterial protein extraction reagent (ThermoFisher), 1 Pierce protease inhibitor tablet (ThermoFisher), 2400 units/mL DNAse I (ThermoFisher), and 50 mg/mL lysozyme (ThermoFisher). The cell pellet was mechanically broken apart with a spatula and further resuspended in wash buffer (PBS, pH 7.4, 20 mM imidazole). The lysate was centrifuged (42,600 × *g* at 4 °C for 15 min) to remove the insoluble fraction and the supernatant was collected by decanting. Metal ion affinity chromatography was used to purify the protein of interest. His-tagged proteins from the soluble fraction were loaded onto HisTrap™ FF Crude Prepacked Columns (GE Healthcare) connected to an ÄKTA™ Pure FPLC system (GE Healthcare). Proteins were eluted from the column using a 0–500 mM imidazole gradient. Imidazole was removed from protein fractions using Amicon Ultra Centrifugal Filters with a 10 kDa cutoff (MilliporeSigma). Protein purity was determined by sodium dodecyl sulfate polyacrylamide gel electrophoresis (SDS-PAGE) and Coomassie staining. Endotoxin content at concentrations used for in vivo experiments was reduced below 1.0 EU/mL, the maximum acceptable dose for pre-clinical drugs[60], using Detoxi-Gel Endotoxin Removing Columns (ThermoFisher). Final endotoxin content was determined with a Pierce LAL Chromogenic Endotoxin Quantitation Kit (ThermoFisher), according to the manufacturer's instructions.

**Size analysis.** Fusion protein MWs were determined under denaturing conditions using SDS-PAGE and protein ladder (BP3602500, ThermoFisher). SEC was used to determine fusion protein MW under native conditions. Briefly, a 250 μL solution of protein in PBS was loaded onto a Superdex™ 20010/30 GL column (GE Healthcare) connected to an ÄKTA™ pure FPLC system (GE Healthcare). Eluted proteins were detected at an absorbance of 280 nm, which was normalized based on maximum signal intensity. Fusion protein MW was calculated by fitting protein elution volume to a calibration curve prepared using protein standard markers (Bio-Rad, GE Healthcare, ThermoFisher) (Supplementary Figure 15). Hydrodynamic diameter of fusion proteins was approximated via DLS on a NanoBrook 90Plus Particle Size Analyzer and BIC Particle Sizing Software (Brookhaven Instruments). Proteins in PBS were filtered and equilibrated to RT before size measurements. The molar concentration of fusion proteins was measured before and after filtration via ultraviolet (UV) absorbance at 280 nm using a NanoDrop™ spectrophotometer (Supplementary Figures 16 and 27–32) (ThermoFisher). Extinction coefficients determined using the ExPASy ProtParam tool were: 2544/M/mm for WT-NL; 6131/M/mm for NL-G3; 6292.5/M/mm for NL-TT-G3; 5639.5/M/mm for GFP-G3; 5788.5/M/mm for GFP-TT-G3; 22,346/M/mm for ChABC-G3; 22,495/M/mm for ChABC-TT-G3. Hydrodynamic diameter ± standard deviation by number-, volume-, intensity-, and in some cases, surface area-weighted size distribution were determined from ten 30 s runs in triplicate or more (Supplementary Figures 33–52).

**G3 fusion partner activity assay.** Molar concentration of fusion proteins was determined using a NanoDrop spectrophotometer, as described above. Standard curves of relative luminescence units versus NL concentration were made by serially diluting NL in white, opaque 96-well microplates (Costar) and adding 50× dilution of stock furimazine (Nano-Glo™ substrate, PRN1120, Promega) in the buffer provided by the manufacturer at a 1:1 volume ratio. Signal was acquired using an open filter (emission = 360–630 nm) and 500 ms integration time on a SpectraMax M3 Multi-Mode microplate reader (Molecular Devices) immediately after the addition of the substrate. To obtain mouse serum, institutional guidelines for the care and use of laboratory animals were strictly followed under a protocol approved by the University of Florida's Institutional Animal Care and Use Committee (IACUC). Bioluminescence produced by NL-G3 and NL-TT-G3 (final NL concentration = 4 nM) in 25% wild-type female C57BL/6 mouse serum was measured over 18 h at 37 °C by combining NL fusions in PBS with 100% serum (1:1 volume) and then supplementing samples with a 50× dilution of stock furimazine (1:1 volume) at the specified time points. Bioluminescence signal was normalized (%) to bioluminescence at the initial time point. Qualitative digital photographic images were taken of blue bioluminescence emitted by WT-NL, NL-G3, and NL-TT-G3 at equal concentrations of enzyme in PBS mixed with 2 μL of stock Nano-Glo™ substrate (Supplementary Figure 17).

GFP fusion concentrations were determined via UV absorbance at 280 nm using a NanoDrop spectrophotometer, as described above. Standard curves of relative fluorescence units (RFU) versus GFP concentration were made by serially diluting GFP in black, clear bottom 96-well microplates (Costar) with excitation = 485 nm and emission = 510 nm using a SpectraMax M3 plate reader. Fluorescence

spectra of GFP fusions were measured at 500 nM GFP with the same settings (Supplementary Figure 18).

ChABC fusion concentrations were determined via UV absorbance at 280 nm using a NanoDrop spectrophotometer, as described above. ChABC activity was confirmed via absorbance at 232 nm in a glass cuvette (NC9469798, ThermoFisher), based on previous methods[61]. Specifically, ChABC-G3 and ChABC-TT-G3 were mixed with their substrate, CS-A (1 mg/mL, C9819 MilliporeSigma), in PBS and their catalytic activity was measured on a SpectraMax M3 plate reader for 15 min with a reading every 1 min. $V_o$ was obtained via linear regression analysis using GraphPad Prism over regions that followed first-order enzyme kinetics. The slope, $V_o$, was then plotted versus the concentration of ChABC.

**Carbohydrate-binding assays.** Fusion protein lactose binding was evaluated using an α-lactose-agarose affinity column connected to an ÄKTA™ pure FPLC system. α-Lactose-agarose resin was purchased from a commercial source (L7634, MilliporeSigma) and packed into a column according to the manufacturer's instructions (GE Healthcare). Approximately 400 μL of 20 μM protein was applied to the column. Unbound protein was removed by washing with 10 column volumes of PBS. Bound protein was eluted using a gradient of 0–100 mM soluble lactose in PBS. Eluted proteins were detected via absorbance at 280 nm, which was normalized based on maximum signal intensity.

Binding of NL and GFP fusions to ASF, laminin, aggrecan, collagen IV, or collagen I adsorbed onto plastic was determined via luminescence and fluorescence, respectively. For NL fusions, a white 96-well microplate was used, while for GFP fusions a black, clear bottom 96-well plate was used. Microplates were coated with 100 μL of 50 μg/mL ASF (A4781, MilliporeSigma), laminin (23-017-015, ThermoFisher), collagen IV (CB-40233, ThermoFisher), aggrecan (A1960, MilliporeSigma), or collagen I (C3867, MilliporeSigma) or PBS control for 2 h at 37 °C. Next, the supernatant was aspirated and wells were washed with PBS followed by blocking with 100 μL of 1% bovine serum albumin (ThermoFisher) for 1 h at RT. Again the supernatant was removed and wells were washed with PBS. Then either 50 μL of NL fusions ([NL] = 20 nM) or 100 μL of GFP fusions ([GFP] = 500 nM) was added to the adsorbed glycoconjugates and incubated for 1 h at RT. As a control, either 50 μL of NL fusions with LacNAc (final [NL] = 20 nM and final [LacNAc] = 10 mM) or 100 μL of GFP fusions (final [GFP] = 500 nM and final [LacNAc] = 10 mM) was added to the adsorbed glycoconjugates and incubated for 1 h at RT. Note: different concentrations of NL and GFP fusions were used to account for differences in the detection limit of each protein (NL ~ 10 pM; GFP ~ 10 nM). Unbound protein was aspirated and wells were washed three times with PBS. Finally, for GFP fusions, bound protein was soaked in 100 μL PBS, and for NL fusions, bound protein was soaked in 50 μL PBS followed by 50 μL furimazine (50× dilution of stock). Fluorescence or bioluminescence emitted was measured for bound GFP or NL, respectively, using a SpectraMax M3 plate reader with excitation = 485 nm and emission = 510 nm (GFP) or an open filter with integration time = 500 ms, similar to methods described above. Binding of a control protein (WT-NL) was also evaluated, as reported in Supplementary Figure 19. Bound GFP and NL concentrations were calculated using a standard curve, as described above.

Micrographs of NL bioluminescence and GFP fluorescence on glass slides coated with laminin (1.2 mg/mL) were obtained by overlaying 50 μL of 5 μM NL or GFP on the slides for 45 min at RT. Laminin was previously adsorbed onto glass in a coffee ring shape from a 2 μL droplet for 1 h at 37 °C. As a control, proteins were mixed with soluble 10 mM LacNAc prior to addition to adsorbed laminin. Unbound GFP was washed off with 100 μL PBS five times before imaging. Unbound NL was washed off with 100 μL PBS, then plates were soaked in 50 μL furimazine (50× dilution of stock) followed by a second 100 μL PBS wash. Images were taken using a Zeiss Axio Observer inverted microscope using a 4,6-diamidino-2-phenylindole (DAPI) filter set for NL (excitation = 380 nm and emission = 450 nm) and GFP filter set for GFP (excitation = 480 nm and emission = 535 nm).

Free GAGs, which lack a hydrophobic protein core, are more challenging to adsorb onto polystyrene surfaces due to their strong hydrophilicity and negative charge[62]. Thus the percentage of enzyme bound to free GAGs was determined using a competition assay with laminin pre-adsorbed onto polystyrene microplates, as described above. One mg/mL CS-A, CS-B, CS-C, and heparin (C9819, C3788, C4384, and H4784, respectively, MilliporeSigma) or PBS control were first mixed with WT-NL, NL-G3, or NL-TT-G3 (final [NL] = 4 nM). Then 50 μL of this solution was added to laminin-coated plates (50 μg/mL laminin) and incubated for 30 min at RT. Unbound enzyme was removed with three PBS washes. Fifty μL furimazine (50× dilution of stock) was added to each well, and immediately thereafter, luminescence produced in each well was quantified using a SpectraMax M3 plate reader using an open filter and 500 ms integration time. Baseline signal produced by WT-NL was subtracted from NL-G3 and NL-TT-G3. Bound enzyme concentrations were calculated using a standard curve, as described above. Percentage of enzyme bound to GAGs was calculated as 1 minus the ratio of [NL] bound to laminin in the GAG group:[NL] bound to laminin in the PBS control.

For competitive inhibition experiments, microplates were coated with ASF, blocked, and washed using the same protocols described above. LacNAc (0–10 mM) was mixed with NL fusions or nanoassemblies ([NL] = 20 nM) at a 1:1 volume ratio and then added to adsorbed ASF for 1 h at RT. Unbound protein was

aspirated and wells were washed four times with PBS before bound NL concentration was determined using a SpectraMax M3 plate reader according to protocols described above. Binding signal was normalized by first subtracting the minimum signal (10 mM LacNAc + protein) and then dividing by maximum signal (0 mM LacNAc + protein). Data were fit via non-linear regression analysis using GraphPad Prism.

Saturation binding curves were generated by adding increasing concentrations of GFP fusions ([GFP] = 0–10 μM) to ASF or laminin pre-adsorbed onto microplates, followed by washing of unbound protein, and finally detection of bound protein, using protocols described above. Data were fit via non-linear regression analysis using GraphPad Prism. Scatchard plots from these data were created by calculating the ratio of bound GFP to free GFP at each GFP concentration for each individual replicate and then plotting the means of these ratios at each GFP concentration versus the mean bound GFP at each concentration. Data for GFP-G3 binding were fit via linear regression and data for GFP-TT-G3 binding were fit via non-linear regression using GraphPad Prism.

WT-G3, NL-G3, and NL-TT-G3 binding affinity for soluble lactose and LacNAc were determined using a tryptophan fluorescence quenching assay, based on previous methods[63]. Specifically, tryptophan fluorescence quenching was detected by mixing in 5 μL increments of $10^{-2}$–$10^4$ μM soluble lactose or LacNAc in water to 500 μL of WT-G3, NL-G3, or NL-TT-G3 ([G3] = 5 μM) in PBS in a quartz cuvette (NC9030411, ThermoFisher) and then measuring tryptophan fluorescence signal at excitation = 280 nm and emission = 335-345 nm with a SpectraMax M3 plate reader. As a control, water was mixed in 5 μL increments to 500 μL of WT-G3, NL-G3, or NL-TT-G3 ([G3] = 5 μM) in PBS in a quartz cuvette. The change in fluorescence signal (ΔRFU) was calculated by subtracting RFU of bound protein (quenched fluorescence signal) from unbound protein (fluorescence signal of protein alone). Representative spectra are shown in Supplementary Figures 20-21. ΔRFU at the lowest and highest concentration of lactose or LacNAc added to G3 proteins were analyzed for statistically significant differences to assess the signal-to-noise ratio for this assay (Supplementary Figure 22). Dissociation constants were calculated via non-linear regression using GraphPad Prism.

Quantitative precipitation of ASF with WT-G3, NL-G3, and NL-TT-G3 was measured by light scattering and absorbance at 420 nm[47,50]. Baseline signal of 7 μM ASF alone had a maximum absorbance of ~0.04. To evaluate assay sensitivity, WT-G3 at different concentrations was mixed with 7 μM ASF and insoluble aggregates that formed were measured by absorbance over a broad wavelength range (Supplementary Figure 23a). Aggregates could be detected at 420 nm when [G3] > 2.5 μM. Based on these observations, absorbance of samples containing 10 μM WT-G3, 7 μM ASF, or 10 μM WT-G3 plus 7 μM ASF in PBS were then measured over a broad wavelength range (Supplementary Figure 23b). WT-G3 and ASF had maximum absorbances of ~0.04, while WT-G3 plus ASF had an absorbance of ~0.4. Based on these observations, WT-G3, NL-G3, or NL-TT-G3 ([G3] = 0–10 μM) was mixed with ASF (7 μM) at a 1:1 volume ratio in a clear 96-well microplate and absorbance at 420 nm was measured immediately thereafter using a SpectraMax M3 plate reader. Additionally, absorbance at 420 nm was collected every 30 s for 10 min to establish a time course for insoluble aggregate formation when WT-G3, NL-G3, or NL-TT-G3 was mixed with ASF (Supplementary Figure 11). Brightfield micrographs and digital photographs were collected as qualitative representation of insoluble aggregate formation when 2.5 or 10 μM G3 was mixed with 7 μM ASF using a Zeiss Axio Observer inverted microscope or digital camera, respectively (Supplementary Figures 9 and 23c). Hydrodynamic diameter ± standard deviation by number-, volume-, intensity-, and in some cases, surface area-weighted size distribution of insoluble aggregates was approximated via DLS using a NanoBrook 90Plus Particle Size Analyzer and BIC Particle Sizing Software from ten 30 s runs in triplicate or more (Brookhaven Instruments) (Supplementary Figures 53-76). Proteins in PBS were filtered and equilibrated to RT before mixing for size measurements. Brightfield and fluorescent micrographs were taken of WT-G3 and GFP-TT-G3 (final [G3] = 10 μM) incubated in PBS with or without 10% fetal bovine serum for 1 h at 37 °C (Supplementary Figure 12). Images were taken on a Zeiss Axio Observer inverted microscope with the same GFP fluorescence filter set described above.

**G3 fusion protein and nanoassembly binding to Jurkat T cells.** For all experiments, Jurkat T cells (Clone E6-1, TIB-152, ATCC) were first expanded in complete media (RPMI 1640 supplemented with 10% heat-inactivated fetal bovine serum, 1% penicillin–streptomycin, L-glutamine 200 mM, 1% HEPES buffer) at 37 °C, 5% CO₂. Cells were then aliquoted (20,000 cells/well) into sterile, clear, tissue culture-treated 96-well microplates. Cells were incubated (final [G3] = 5 μM) for 4 h at 37 °C and then transferred to a V-bottom 96-well microplate and collected (400 × g for 5 min) using a Jouan CR3i Multifunction Centrifuge equipped with a microplate rotor. The supernatant was carefully pipetted from the wells without disrupting the cell pellet. Cells were resuspended in 50 μL PBS for NL fusions or 100 μL PBS for GFP fusions. Cells were transferred to either a white, opaque 96-well microplate for detection of NL or black, clear bottom 96-well microplate for detection of GFP fusions. Fluorescence from GFP fusions bound to cells was measured directly on a SpectraMax M3 plate reader (excitation = 485 nm, emission = 510 nm). For cells treated with NL fusions, 50 μL furimazine (50× dilution of stock) was added to each well and bioluminescence produced in each well was quantified immediately using a SpectraMax M3 plate reader using an open filter and 500 ms integration time. Bound NL or GFP concentrations were

calculated using a standard curve, as described above. Micrographs of NL and GFP localized to the surface of Jurkat T cells, using protocols described above, were taken with a Zeiss Axio Observer inverted microscope with DAPI and GFP fluorescent filter sets described before.

**Changes in Jurkat T cell phenotype and function**. Extracellular activity of WT-G3, NL-G3, and NL-TT-G3 was characterized using Jurkat T cells. For all experiments, cells were first expanded in complete Jurkat T cell media (RPMI 1640 supplemented with 10% heat-inactivated fetal bovine serum, 1% penicillin–streptomycin, L-glutamine 200 mM, 1% HEPES buffer) at 37 °C, 5% $CO_2$. Cells were then aliquoted (20,000 cells/well) into sterile, clear, tissue culture-treated 96-well microplates. To evaluate agglutination, cells were incubated with WT-G3, NL-G3, or NL-TT-G3 (final [G3] = 5 µM) and then imaged intermittently over 4 h using a Zeiss Axio Observer inverted microscope. To determine changes in cell metabolic activity, Jurkat T cells were incubated with WT-G3, NL-G3, or NL-TT-G3 ([G3] = 5 µM) for 4 h followed by incubation with 20 µL/well CellTiter-Blue reagent (PR-G8080, Promega) for 2 h[63]. Fluorescence produced in each well was then measured using a SpectraMax M3 plate reader (excitation = 560 nm, emission = 590 nm). Background fluorescence of culture media plus CellTiter-Blue Reagent was subtracted from sample fluorescence. Finally, relative metabolic activity was reported as fluorescence of cells incubated with WT-G3, NL-G3, or NL-TT-G3 normalized to the fluorescence of untreated cells (PBS).

Jurkat T cell phosphatidylserine exposure and PI permeability were evaluated after treating cells with PBS (negative control), WT-G3 (positive control), NL-G3, and NL-TT-G3. Protocols for this experiment were adapted from prior reports[64], using a BD Annexin V: fluorescein isothiocyanate (FITC) Apoptosis Detection Kit I (BD556547, ThermoFisher). Briefly, 500 µL of $2 \times 10^6$ cell/mL was mixed with 500 µL of 10 µM G3 in PBS in a 15 mL conical tube. Cells were then incubated for 4 h at 37 °C in a water bath before being cooled on ice for 10 min. 200 mM ice-cold lactose was added to the cells to remove bound G3, followed by 10 mL of 100 mM lactose and centrifugation at $413 \times g$ for 4 min on a Centrifuge 5804R (Eppendorf). Supernatant was removed thereafter and cells were resuspended in 1 mL 1× Annexin V Binding Buffer (BD556547, ThermoFisher). 100 µL of the cells (~$10^5$ cells) were transferred to a 1.5 mL microtube and mixed with 5 µL FITC–Annexin V and 5 µL PI (BD556547, ThermoFisher). Cells were vortexed gently and incubated for 15 min at RT in the dark. 100 µL of 1× Annexin V Binding Buffer was added to the tube and then 100 µL was transferred to a glass-bottom microwell (NC9069930, ThermoFisher) for brightfield and fluorescence imaging. Images were taken on a Zeiss Axio Observer inverted microscope with FITC (excitation = 470/40 nm and emission = 525/50 nm) and rhodamine (excitation = 546/12 nm and emission = 575–640 nm) filter set. Individual and overlaid images are available at two magnifications in Supplementary Figures 7-8.

To quantify the amount of IL-2 secreted by Jurkat T cells treated with WT-G3, NL-G3, or NL-TT-G3, we expanded and aliquoted cells at the same cell density as above into sterile, clear tissue culture-treated 96-well microplates. Cells were incubated with PBS, WT-G3 in PBS, NL-G3 in PBS, NL-TT-G3 in PBS, WT-G3 + 25 mM lactose in PBS, NL-G3 + 25 mM lactose in PBS or NL-TT-G3 + 25 mM lactose in PBS (final [G3] = 2.5 µM) for 24 h at 37 °C. G3 concentration and incubation time were chosen based on previous reports of WT-G3 induced secretion of IL-2 by Jurkat T cells[46]. Cells were then centrifuged, as described above, and the supernatant was collected and analyzed for human IL-2 using a commercially available solid-phase ELISA kit (Quantikine Human IL-2 Immunoassay, D2050, R&D systems), according to the manufacturer's instructions. IL-2 concentration was calculated from the standard curve shown in Supplementary Figure 24.

**In vivo imaging and pharmacokinetics**. In all in vivo imaging and pharmacokinetics experiments, institutional guidelines for the care and use of laboratory animals were strictly followed under a protocol approved by the University of Florida's IACUC. Each injection site had an independent cohort (N = 5) of 8-week-old, female wild-type C57BL/6 mice (The Jackson Laboratory). Prior to protein injection, mice were anesthetized with isoflurane and treated with hair removal cream at the injection site. While anesthetized, mice received a single injection of 40 µL WT-NL, NL-G3, or NL-TT-G3 ([NL] = 3.27 µM for all formulations) in sterile PBS subcutaneously into the hock or scruff or IM into the caudal thigh muscle. Immediately thereafter, mice received a 40 µL injection of furimazine (50× dilution of stock) in sterile PBS. Mice were then anatomically positioned with injected tissue facing the charge-coupled device camera of an IVIS Spectrum In Vivo Imaging System (PerkinElmer) and whole-body bioluminescent images were taken (note: in all experiments this is referred to as t = 0). Every 24 h thereafter, mice were anesthetized, and substrate was again injected into the hock, scruff, or thigh. Images were captured immediately thereafter as described for t = 0. Bioluminescent images were captured using an open emission filter, subject size 1.5 cm, 1 s exposure time, field-of-view B (6.6 cm), medium binning (factor of 8) resolution, and a 1 F/Stop aperture. Relative light intensities, corresponding with local bioluminescence, were represented by a pseudo color scale ranging from violet (least intense) to red (most intense). Signal produced as color also represented photons, which was then converted to photon flux (photons/s) within a circular region of interest (ROI) using the Living Image analysis software. For each injection site, the size of the ROI was manually drawn out to the perimeter of the

bioluminescence signal produced at t = 0. Color scale limits were adjusted to min = 1e6 and max = 4e8 manually for all images, and color scale bars are presented in log scale. Subsequently, all bioluminescent images were analyzed using the same size ROI to normalize for background signal and accurately quantify the decay in bioluminescence over time. Data (photon flux versus days) were analyzed with GraphPad Prism software using nonlinear regression to curve fit a one-phase exponential decay and calculate a bioluminescence half-life. For samples in which bioluminescence above baseline was only detectable in tissue at t = 0, half-life could not be detected because insufficient data points were available for non-linear regression curve fitting. Bioluminescence data for each mouse at t = 0 are shown in Supplementary Figure 25.

To detect the presence of NL in circulation, blood was drawn at 6 h and then daily thereafter from a cohort (N = 5) of 8-week-old, female wild-type C57BL/6 mice that received a 40 µL subcutaneous injection of NL-G3 or NL-TT-G3 ([NL] = 3.27 µM for both formulations) into the hock. At each time point, ~10 µL of blood was drawn from the tail vein using an 18-gauge needle. Blood samples were immediately mixed with 1 µL of 50 mM EDTA (anticoagulant), according to established methods[65]. Five µL of blood was mixed with 50× dilution of furimazine (1:10 volume ratio) in a white, opaque 96-well microplate. Luminescence was immediately measured on a SpectraMax M3 plate reader. Background luminescence produced by the blood of mice that did not receive NL-G3 or NL-TT-G3 was subtracted from measured luminescence. NL concentration in blood was then calculated from luminescence using a standard curve, as described above. The amount of NL detected in blood was reported as the percentage of the total protein by mass injected into the hock.

**Immunogenicity assay**. In all immunogenicity studies, institutional guidelines for the care and use of laboratory animals were strictly followed under a protocol approved by the University of Florida's IACUC. Eight-week-old, female wild-type C57BL/6 mice (N = 5) received scruff injections of 100 µL TT-GFP (control lacking G3), WT-NL, NL-G3, or NL-TT-G3 ([NL] or [GFP] = 1 µM for all formulations) in sterile PBS, based on previous methods[66]. For positive control, cohorts of mice were injected with WT-NL or TT-GFP ([NL] or [GFP] = 1 µM) emulsified in TiterMax Gold™ Adjuvant (T2684, MilliporeSigma). Blood was drawn from the submandibular maxillary vein every 2 weeks for 8 weeks. Immediately after blood was collected, sera were isolated via centrifugation and frozen until analysis. At 4 weeks, mice received a second injection of 50 µL of 1 µM protein in PBS or emulsified adjuvant, respectively. Anti-protein antibodies raised in mice were detected with peroxidase-conjugated goat anti-mouse IgG (NC9731556, Thermo-Fisher) at 450 nm absorbance in ELISA microplates coated with 1 µg/mL protein in PBS or with PBS alone (control), based on previous methods[67].

**Collagenase digestion**. To characterize collagenase-mediated degradation, WT-G3, NL-G3, or NL-TT-G3 were treated with collagenase from *Clostridium histolyticum* (C7657, MilliporeSigma) in vitro, based on previous methods[36]. Specifically, collagenase was added to WT-G3, NL-G3, or NL-TT-G3 in PBS at a 3:1 mass ratio. The samples were then incubated at 37 °C for 2 h for SDS-PAGE analysis or 18 h for SEC (Supplementary Figures 26 and 13, respectively).

**Statistical analysis**. All experimental and control groups were n = 3 for enzyme and GFP activity and were reported as average ± standard deviation. DLS measurements were run at minimum in triplicate, and the data were reported as average ± standard deviation. All experimental and control groups were n = 3 for glycoconjugate, competitive inhibition, and saturation binding data. Dissociation constants for GFP-G3 from Scatchard curves were calculated by linear regression using the GraphPad Prism software. All experimental and control groups were n = 3 for tryptophan fluorescence quenching experiments, and the data were reported as average ± standard deviation. Dissociation constants from tryptophan fluorescence quenching experiments were calculated by non-linear regression using the GraphPad Prism software. All experimental and control groups were n = 4 for Jurkat T cell quantitative binding, metabolic activity, and IL-2 expression experiments, and the data were reported as average ± standard deviation. All experimental and control groups were n = 3 for ASF precipitation experiments, and the data were reported as average ± standard deviation. All experimental and control groups were n = 5 for animal experiments, and the data were reported as average ± standard deviation. Data obtained from DLS and protein concentration measurements are technical replicates. In all other cases, n refers to the number of biological replicates. To calculate bioluminescence half-life, non-linear regression with GraphPad Prism software was used to curve fit a one-phase exponential decay. $p$ = 0.0472 for hock:NL-G3 half-life versus hock:NL-TT-G3 half-life and $p$ = 0.0219 for scruff:NL-G3 half-life versus scruff:NL-TT-G3 half-life. For all other studies, $p$ values are indicated as follows: $*p < 0.05$, $**p < 0.01$, $***p < 0.001$, $****p < 0.0001$, and n.s. is no significant difference or $p \geq 0.05$. Statistical differences between groups were analyzed using two-tailed Student's $t$ test (two groups) or analysis of variance with Tukey's post hoc (multiple groups) in the GraphPad Prism software.

In GAG-binding assays, the $t$ values are $t(4) = 8.347, 47.520, 38.580$, and $11.680$ for CS-A, CS-B, CS-C, and heparin, respectively. For comparison of dissociation constants from tryptophan fluorescence quenching experiments with LacNAC and lactose, the $F$ values are $F(2,6) = 54.980$ and $22.770$, respectively. For comparison

of minimum versus max ΔRFU signal in tryptophan fluorescence quenching experiments, the $t$ values are $t(4) = 18.560$, $15.100$, and $31.860$ for WT-G3, NL-G3, and NL-TT-G3, respectively, treated with lactose. For comparison of minimum versus maximum ΔRFU signal in tryptophan fluorescence quenching experiments, the $t$ values are $t(4) = 18.550$, $27.090$, and $38.780$ for WT-G3, NL-G3, and NL-TT-G3, respectively, treated with LacNAc. In Jurkat T cell quantitative binding assays, the $t$ values are $t(6) = 8.443$ and $8.679$ for NL fusions and GFP fusions, respectively. In the Jurkat T cell metabolic activity experiment, the $F$ value is $F(2,9) = 105.800$. In the Jurkat T cell IL-2 expression assay, the $F$ value is $F(6,21) = 35.710$. For comparison of in vivo half-life, the $F$ value is $F(1,23) = 16.510$. For comparison of catalytic activity at initial time points in vivo, the $F$ values are $F(2,12) = 2.496$, $15.100$, and $9.323$ for injections into the hock, scruff, and thigh, respectively.

## Data availability

The data that support the findings of this study are available from the corresponding author upon reasonable request. Genetic and amino acid sequences of NL-G3, NL-TT-G3, GFP-G3, GFP-TT-G3, ChABC-G3, ChABC-TT-G3, and TT-GFP were deposited in GenBank with the accession codes MH920 252, MH920 253, MH920 254, MH920 255, MH920 256, MH920 257, and MH920 258, respectively.

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

## Acknowledgements

This research was supported by the National Institutes of Health (NIBIB, 5-R03-EB019684-02; NIAID, 1-R01-AI133623 01; NIDCR, 1-R01-DE027301-01; NIDDK, 5-R01-DK098589-04). The content is solely the responsibility of the authors and does not necessarily represent the official views of the National Institutes of Health. We thank Dr. Carlos Rinaldi and Eric Fuller for access and time using their dynamic light scattering instrument.

## Author contributions

G.A.H. conceived the project in collaboration with B.G.K. G.A.H. designed experiments, analyzed data, and contributed to the writing and editing of the manuscript. S.A.F. conducted experiments, analyzed data, designed figures, and contributed to writing and editing of the manuscript. E.B.-S. and A.R. established protocols for several experiments and also contributed to data collection and analysis. E.B.-S. and S.L.F. contributed to in vivo imaging experiments. M.M.F. contributed to cloning, immunogenicity, and IL-2 secretion experiments. D.T.S. contributed to expression, purification, and characterization of ChABC-G3 and ChABC-TT-G3.

## Additional information

**Competing interests:** A pending patent has been filed by the University of Florida related to materials reported in this manuscript that lists G.A.H., B.G.K., S.A.F., E.B.-S., A.R., M.M.F., and S.L.F. as inventors. Publication No. WO2018067660A1; Publication date 04/12/2018. A provisional patent has been filed by the University of Florida related to materials reported in this manuscript that lists G.A.H., B.G.K., D.T.S., and S.A.F. as inventors. Serial No. 62/751,146; Filing date 10/26/2018.

