## [Peer Review File · Nature Communications]

Reviewers' Comments:

Reviewer #1:

Remarks to the Author:

This is an elegant and careful manuscript that demonstrates trimeric fusions of enzymes with G3 domains bind glycans in vitro and lead to significantly improved half-lives of enzymes in vivo following subcutaneous or intramuscular injection. The conclusions are well supported by the experimental results provided. This is a novel approach that takes advantage of a native ligand binding interaction, and further improves it with multivalency. Furthermore, the authors were careful to assess potential cytotoxicity and immunogenicity and also offer data on the potential mechanism of increased longevity of the trimer fusion. This is an exciting platform that could be applied to a wide variety of injected therapeutic proteins with significant benefit over existing technologies or unmodified proteins. The methods and statistics are appropriate and fully detailed.

There are a few minor points that should be addressed, listed below.

1. A little more background on G3 would be helpful. Is it a human protein, or from some other species? What is its native purpose (i.e. why does it bind glycans)?
2. The immunogenicity assay is important to do, but it is also difficult to guess how frequent dosing ("vaccinating") should be since ultimately that would depend on the therapeutic protein, disease state, etc. Context is needed for 2 doses 1 month apart. While this is a typical dosing schedule for a vaccine, it is not necessarily typical for a protein therapeutic. Do pegylated proteins induce anti-PEG antibodies on dosing schedule like this, or are many more doses required?
3. There is literature describing fusion of collagen binding domains and fibrin binding domains to growth factors for immobilization in ECM. Though the applications for this are typically in tissue engineering and there is not multimerization, the concept is related and should have been discussed alongside the antibody binding strategies in the introduction. 2 examples: Kitajima, Takashi, et al. "A chimeric epidermal growth factor with fibrin affinity promotes repair of injured keratinocyte sheets." *Acta biomaterialia* 5.7 (2009): 2623-2632. Kitajima, Takashi, Hiroshi Terai, and Yoshihiro Ito. "A fusion protein of hepatocyte growth factor for immobilization to collagen." *Biomaterials* 28.11 (2007): 1989-1997.
4. Supp Fig 1 is so narrowly cropped there is no sense of the purity of the proteins.
5. Is DLS data in Fig 1g displayed as intensity signal, or a manipulated signal such as volume or number? To make the claim there are no aggregates forming the intensity signal should be shown and the x axis should be extended to 1 micron.
6. In Fig 3E, metabolic activity was measured not viability. The y axis should reflect the actual measurement.

Reviewer #2:

Remarks to the Author:

Farad et al. provide their new work on a protein fusion construct comprising a luciferase enzyme and a targeting ligand – the known carbohydrate-binding protein, G3. The Luc-G3 fusion protein is shown to bind soluble lactose in solution, glycans on plastic surfaces, glycan-decorated cell cultures, and in vivo tissue sites upon injection into rodents. The authors propose the design as a better way to ensure tissue-site binding and retention for enzyme therapeutics.

While the breadth and depth of the assays, assessments and data is sufficient, two primary issues cloud this reviewer's enthusiasm for Nature consideration. First, the actual novelty and relevance of the fusion protein design to therapeutic use is limited. Their stated goal of "anchoring enzymes to tissues via G3-mediated glycan binding (to) be broadly useful for improving local pharmacokinetics of various existing and emerging enzyme drugs" is attractive, but not shown to be an improvement over any other known ligand-targeted strategy. Certainly, targeted enzyme conjugates are long known and not compared, and their proof of concept shown using nanoLuc is limiting in asserting these broad, general therapeutic claims. Other enzyme-affinity fusions are

precedents. Even 1980's papers describe carboxypeptidase-dextran conjugates with high circulating half-life and intrinsic bioactivity. Second, a sufficient amount of technical questions remain about what is actually measured, what it means and how it impacts the significance of the findings for the purported goal of anchoring enzymes on tissue surfaces. Specific issues to consider for improvements:

1. The two enzymes mentioned as therapeutic justifications have molecular weights exceeding 40kDa each, while nanoLuc is only about 18kDa, substantially smaller and, unlike the therapeutic comparators, specifically engineered for stability. Use of nanoLuc is interesting for proof of concept but should be extended to more therapeutic relevance in this publication for significance and general utility by using larger enzymes similar to the therapeutics.
2. nanoLuc™ is actually a product tradename of Promega, not the name of a protein. This needs to be described and changed in the manuscript.
3. Pre-loading the WT G3 and G3-Luc conjugates with lactose then running them down an affinity column or over protein-coated surfaces or in cell culture assays should have been used to assess non-specific G3 binding effects.
4. Suppl. Fig 1: Why not run WT G3 on this gel for comparison?
5. Figure 4 and methods: the luciferase substrate emits in the blue spectrum - not ideal for in vivo monitoring due to overlap with other biological components:
6. Figure d, e, f: how reliable are these optical emission values (y-axis, $\sim 10^9$) for quantitative comparison since the enzyme and luciferase substrate injections are highly variable at tissue sites and the substrate has some PBS solubility issues as well? How are these y-axis time-zero Photon flux values so tight and similar for Figures 4d, e, f across all animals and tissue injection sites when they are usually highly variable?
7. Page 17, endotoxin: LPS values are not reported anywhere?
8. Page 18: Nano-Glo substrate, furimazine, excitation and emission wavelengths should be reported here despite use of an "open filter".
9. Page 18: "Catalytic activity of 4 nM WT-NL, NL-G3, or NL-TT-G3 in wild-type female C57BL/6 mouse serum...": complete undiluted serum?
10. Page 19: "... we instead used the molar concentration of glycoprotein/proteoglycan adsorbed onto the plate as the total ligand concentration..." how was this adsorbed amount determined on surfaces?
11. P. 20: "black, glass-bottom 96-well plates...": since typical glass is not able to transmit UV wavelengths effectively below 360nm, was this UV-grade optical glass or quartz as required for such assays? What is its optical transmissivity at 280nm?
12. P. 20: Trp fluorescence: please show sample excitation and emission vs wavelength curves in the supplemental information for before and after lactose binding and how these were used to calculate (Δ)RFU as reported.
13. P. 20, last line: please provide the assay LOD, sensitivity, and Δ in optical signal at 420nm for aggregation? What (Δ amplitude) difference measured is actually significant for light scattering? Please show sample curves for this difference in the Supplemental figures.
14. P. 21: "Cell-Titre Blue": this is not an apoptosis-specific assay - this is a general cell viability/toxicity assay. Apoptosis is not shown.
15. P. 21: since G3 is a known T-cell activator, relevant changes in cytokine production should also be shown.
16. P. 21: why different time points for cell surface binding assay versus cell viability assay?
17. P. 24: ELISA assay for anti-G3 antibodies in murine plasma: what's the positive control to show that this ELISA can detect anything specific? The assay should use spiked controls for a standard curve using known G3 additions to murine plasma. There's no proof provided that this assay can work at all. Please show an ELISA standard curve in the supplemental figures. Only two injections of antigen? How is this justified?

Reviewer #3:

Remarks to the Author:

Manuscript: NCHEMB-A081103318

Authors: Shaheen A. Farhadi, Evelyn Bracho-Sanchez, Margaret M. Fettis, Sabrina L. Freeman, Antonietta Restuccia, Benjamin G. Keselowsky, and Gregory A. Hudalla*

Title: Locally anchoring enzymes to tissues via extracellular glycan recognition

Comments for the authors:

The manuscript describes the generation of enzyme/galectin-3 (Gal-3) fusion proteins and their utilization for localizing the enzyme activity to glycoconjugates in the ECM (glycoproteins, glycosaminoglycans GAGs) and cell surfaces (Jurkat T cells). The authors choose the enzyme nanoluciferase (NL) for fusion to the N-terminus of Gal-3. A special protein construct renders self-assembled trimeric Gal-3/NL protein structures.

After protein chemical characterization of these Gal-3 fusion proteins, the authors undertake a series of experiments to characterize their binding affinities to lactose, glycoproteins, of the ECM, and glycosaminoglycans GAGs). Most importantly, the authors demonstrate in a mouse model prolonged bioluminescence in the tissue, with the trimeric NL/Gal-3 constructs being most efficient. Further experiments show no immunogenic effect, no accumulation in blood serum as well as susceptibility to proteolytic cleavage in the N-terminal domain of Gal-3 by collagenase. In summary, the author suggest enzyme/Gal-3 fusion as general strategy to anchor enzymes at injection sites that may be useful for improving local pharmacokinetics of enzyme drugs.

The manuscript is well written and presentation and discussion of the results is in general concise. The concept of enzyme/galectin fusion is not new and has been formerly reported by Qasba et al. (Biochem. Biophys. Res. Commun. 2010, 394, 679-684) and Chammas et al. (Histochem. Cytochem. 2007, 55, 1015-1026), both references are not mentioned in the manuscript. The authors do not explain why they chose Gal-3 as lectin component – other lectins are also useful to prolong enzyme activities in tissues. The impact of these protein constructs may also lie in the fact that Gal-3 is a biomarker for tumor progression and angiogenesis.

Gal-3 is very well characterized in terms of their glycan ligands on glycoproteins. This is not sufficiently addressed in the manuscript and parts of glycan binding characterization suffers from some weak points making subsequent conclusions questionable.

Page 8, first paragraph:

The authors' statement: "Significantly more enzyme bound to adsorbed glycoconjugates when formulated as NL-TT-G3 nanoassemblies versus monomeric NL-G3 fusion proteins on a G3 molar basis (Fig. 1i, j, Fig. 2a, Supplementary Fig. 3), demonstrating that NL-TT-G3 had higher apparent binding affinity for surface adsorbed glycoconjugates than NL-G3. We estimated NL-G3 and NL-TT-G3 relative binding affinity for different glycoconjugates by assuming that the carbohydrate concentration on the surface was equal to the glycoconjugate concentration coated onto the plate."

Higher amounts of adsorbed enzyme activity does not necessarily indicate higher lectin affinity. Theoretically, binding of only one Gal-3 moiety of NL-TT-G3 on the adsorbed glycoprotein would result in three-fold higher enzyme activity. Enzyme activity read-out is not suitable to assess lectin affinity characteristics. What is meant with [NL] added in Fig. 2a (x-axis). Is it a normalized value in terms of NL concentration of the constructs (NL-TT-G3 with three NLs and NL-G3 with one NL)? In Fig 2a binding sites of laminin seem to be already saturated at 1 nM NL. Binding curves with binding signal over concentration of fusion construct with readouts directly associated with Gal-3 binding (anti-Gal-3 antibodies, anti-HisTag) are more suitable for demonstrating binding efficiency. The assumption that glycan concentration equals the concentration of adsorbed glycoproteins is not correct. Glycoconjugates are known to be only partially modified with glycans. Therefore, the

presented determination and calculation of dissociation constant (Fig. 2b) is highly questionable. Instead, competitive binding assays giving IC50 values for competitive glycan ligands (lactose, LacNAc) should be applied. Binding strength of the Gal-3 fusion proteins should be then compared by IC50 values derived from competitive binding curves with soluble sugars/glycans with LacNAc (Galbeta1-4GlcNAc) as competitive soluble glycan ligand.

Page 7, Fig 2c:

The experiment should exclude whether Gal-3 constructs are bound to GAGs by ionic interaction. A suitable control should be included. What are the pI values of the Gal-3 constructs?

Page 9 and Fig.2d:

The conclusions are solely based on lactose (Fig 2d), which is not a constituent of glycans presented on the glycoconjugates used in this study. From this point of view, binding experiments with free lactose are not meaningful. It is well known from literature that LacNAc and poly-LacNAc, e.g. on laminin, are the glycans recognized by Gal-3. Conclusions from lactose binding studies of Gal-3 can therefore not be valid for glycoproteins, which vary in their content of LacNAc and poly-LacNAc.

Apparent binding constants could be derived from binding experiments with immobilized asialofetuin (ASF), which is standard glycoprotein for galectin binding experiments with known N-glycan composition.

The following points should be addressed in the 'Methods' part:

- What is the origin of the lactose affinity chromatography column?
- Show calibration curves for NL activity over NL concentration (supplementary data).
- Carbohydrate binding assays: The calculation is highly questionable.

The assumption that glycan concentration equals the glycoconjugate concentration is not valid. Glycans are only part of the glycoconjugates. The ligand concentration is much smaller than assumed.

Reviewer #1

1. A little more background on G3 would be helpful. Is it a human protein, or from some other species? What is its native purpose (i.e. why does it bind glycans?)?

We appreciate this suggestion from the reviewer. We have added new text to the third paragraph of the introduction that explains the native functions of G3 and points readers to a relevant review article.

2. The immunogenicity assay is important to do, but it is also difficult to guess how frequent dosing ("vaccinating") should be since ultimately that would depend on the therapeutic protein, disease state, etc. Context is needed for 2 doses 1 month apart. While this is a typical dosing schedule for a vaccine, it is not necessarily typical for a protein therapeutic. Do pegylated proteins induce anti-PEG antibodies on dosing schedule like this, or are many more doses required?

We agree with the reviewer on this point, as does reviewer #2. This is a very difficult model to develop because it will in fact depend highly on the protein, the disease state, the dose, the injection site, and dosing frequency, and should be evaluated rigorously on a case-by-case basis. We initially chose 2 doses 1 month apart because this is sufficient to elicit high anti-protein serum IgG titers when the protein is administered with an adjuvant, as demonstrated with our positive control data for both nanoluciferase and TT-GFP emulsified in TiterMax. We believe that this experimental design allows us to ask the fundamental question, does prolonging local retention increase host generation of anti-protein antibodies reactive against G3-fusion proteins, to which our data suggest that the answer is no. However, we fully recognize that this schedule may not be therapeutically relevant. Rather, we envision that proteins will need to be delivered more frequently and possibly at higher doses to fully evaluate immunogenicity.

To address this critique, we have (1) edited the language in the text to more clearly outline our experimental design and motivation for these studies, (2) moved these data to the supplemental materials to de-emphasize their significance in this study (now Fig. S14), and (3) removed text in the abstract and conclusion attesting to the lack of immunogenicity of these fusion proteins and assemblies.

*3. There is literature describing fusion of collagen binding domains and fibrin binding domains to growth factors for immobilization in ECM. Though the applications for this are typically in tissue engineering and there is not multimerization, the concept is related and should have been discussed alongside the antibody binding strategies in the introduction. 2 examples: Kitajima, Takashi, et al. "A chimeric epidermal growth factor with fibrin affinity promotes repair of injured keratinocyte sheets." *Acta biomaterialia* 5.7 (2009): 2623-2632. Kitajima, Takashi, Hiroshi Terai, and Yoshihiro Ito. "A fusion protein of hepatocyte growth factor for immobilization to collagen." *Biomaterials* 28.11 (2007): 1989-1997.*

We appreciate this recommendation by the reviewer, which was also raised in other contexts by reviewers #2 and 3. To comply with editorial guidelines, we had to make decisions regarding references to include and exclude, which unfortunately led us to neglect some previous reports that are relevant to our work. We initially chose to forego these references because they are primarily in vitro examples of growth factor retention. Nonetheless, they are relevant to our report. To address this critique, we have included these references, as well as related citations to “superaffinity” peptides reported by Hubbell and colleagues, as a motivation for our work in the first sentence of the fourth paragraph of the introduction.

4. Supp Fig 1 is so narrowly cropped there is no sense of the purity of the proteins.

We agree with the reviewer on this point, and have added revised SDS-PAGE gels to the supplemental materials (Fig. S1) which present a much broader molecular weight range for G3 fusion proteins included in this report. Likewise, the size exclusion chromatography data presented in figure 2 provide a reasonable indication of the purity of these proteins, since they cover the spectrum of ~20-600 kDa.

5. Is DLS data in Fig 1g displayed as intensity signal, or a manipulated signal such as volume or number? To make the claim there are no aggregates forming the intensity signal should be shown and the x axis should be extended to 1 micron.

We appreciate these comments. The DLS data are reported as number, which is more common for proteins due to their relatively small size. To address this critique, DLS data presented in figure 2 has been modified to present the range of 1 nm – 1 micron. Likewise, we have included the raw intensity, number, and volume data, as well as the fitting of these data, to the supplemental information file. Finally, we have added data to the supplemental materials reporting the protein concentration before and after filtration with a 0.2 micron syringe filter, which demonstrates that protein concentration does not change considerably (Fig. S16). Collectively, these data support our conclusion that some aggregation does occur but the propensity for aggregation is very low.

6. In Fig 3E, metabolic activity was measured not viability. The y axis should reflect the actual measurement.

We appreciate the reviewer’s attention to detail and have edited the y-axis in this figure, now 4e, to reflect this point.

Reviewer #2

(1) First, the actual novelty and relevance of the fusion protein design to therapeutic use is limited. Their stated goal of “anchoring enzymes to tissues via G3-mediated glycan binding (to) be broadly useful for improving local pharmacokinetics of various existing and emerging enzyme drugs” is attractive, but not shown to be an improvement over any other known ligand-targeted strategy. Certainly, targeted enzyme conjugates are long known and not compared, and their proof of concept shown using nanoLuc is limiting in asserting these broad, general therapeutic claims. Other enzyme-affinity fusions are precedents. Even 1980's papers describe carboxypeptidase-dextran conjugates with high circulating half-life and intrinsic bioactivity.

We appreciate this assessment by the reviewer, and while we agree on some points, we disagree on others. In this communication, we report the development and detailed characterization of a new platform for localized enzyme retention within tissues. Demonstrating therapeutic use of this platform is beyond the scope of this communication but will soon follow, and will likely involve the chondroitinase ABC-G3 constructs included in the revised manuscript, as well as others currently being developed. Beyond exploiting carbohydrates as universal anchoring points found in various tissues, the novelty of this system is inherent in the tunability of local enzyme retention it affords, here enabled by assembling enzyme-G3 fusions into a multivalent architecture. As suggested by demonstrations within this report, such tunability can provide control of local enzyme activity half-life, which we expect will be advantageous for optimizing local pharmacodynamics of established and emerging enzyme drugs.

Regarding more specific criticisms, certain targeted enzyme strategies have proven effective in specific disease contexts, yet are not necessarily broadly applicable. For example, mannose receptor targeting enabled by modifying enzymes with glycans greatly improved therapeutic efficacy of enzyme replacement therapies for Gaucher disease, yet are modestly effective for treatment of Fabry disease because they tend to mediate enzyme accumulation at off-target sites, such as the liver and spleen. Likewise, antibody-enzyme conjugates that target tumors have proven effective in pre-clinical models, yet clinical effectiveness has been limited, as discussed in the introduction of our manuscript with regard to carboxypeptidase G2 and its unfavorable immunogenicity (reviewed in Sharma, S.K. *Adv. Drug Del. Rev.* 2017). Thus, we did not include a ligand-targeted enzyme as a comparator because there is no gold-standard to benchmark against. Likewise, carboxypeptidase-dextran conjugates are only marginally relevant because, as outlined schematically in figure 1 and demonstrated in figures 5-6, our objectives are not to improve enzyme half-life in circulation. Rather, our approach is explicitly designed to retain enzymes locally in proximity to minimally-invasive injection sites. Furthermore, dextran has been shown to be immunogenic when conjugated to proteins (Seppala, *I Eur J Immunol* 1985), which has greatly limited its clinical use for improving protein half-life and bioactivity. We note that to comply with editorial guidelines, we initially limited the references in the introduction of the initial submission to emphasize modified enzymes that have entered clinical trials or are currently approved for clinical use. However, we appreciate that there is a rich, well-established field of modified enzymes that predates our work. Thus, to address this critique, we have added additional text and references to the second paragraph of the introduction that is intended to exemplify the strengths and weaknesses of the broader spectrum of modified enzymes that have been reported to date, which we use to more clearly motivate our approach

and enzyme fusion design. We also reiterate here, as well as in edited text in the manuscript, that our approach is not a ligand-targeting approach, in that we are not introducing our constructs into circulation and expecting them to home to a particular tissue site. Rather, our approach is intended to circumvent practical challenges of ligand-targeted delivery – namely, the need to cross endothelial barriers and potential for renal clearance – by enabling tissue site-specific delivery of enzymes via minimally-invasive injections.

Specific points:

1. The two enzymes mentioned as therapeutic justifications have molecular weights exceeding 40kDa each, while nanoLuc is only about 18kDa, substantially smaller and, unlike the therapeutic comparators, specifically engineered for stability. Use of nanoLuc is interesting for proof of concept but should be extended to more therapeutic relevance in this publication for significance and general utility by using larger enzymes similar to the therapeutics.

We greatly appreciate this suggestion. To address this critique, we have included characterization of 4 additional galectin-3 fusion constructs in figure 2 of the revised manuscript: GFP-G3, GFP-TT-G3, Chondroitinase ABC (ChABC)-G3 and ChABC-TT-G3. The GFP constructs do demonstrate that proteins larger than nanoluc can be fused to galectin-3 and assembled into supramolecular structures without loss of activity, but were primarily useful for addressing concerns that enzyme activity is not an accurate read-out of extent of galectin fusion binding to glycoproteins and proteoglycans. The chondroitinase ABC constructs, on the other hand, demonstrate that an enzyme that is (1) much larger than the therapeutics discussed in the paper, which (2) is considerably less stable than nanoluc and GFP, can be fused to galectin-3 without appreciable loss of activity.

2. nanoLuc™ is actually a product tradename of Promega, not the name of a protein. This needs to be described and changed in the manuscript.

We apologize for this oversight and appreciate the reviewer's attention to detail. We have edited the text in the last paragraph of the introduction, as well as in the methods under the heading, "**Cloning, expression, and purification of recombinant proteins**", to correct this error. We have also added a relevant reference to the introduction and methods.

3. Pre-loading the WT G3 and G3-Luc conjugates with lactose then running them down an affinity column or over protein-coated surfaces or in cell culture assays should have been used to assess non-specific G3 binding effects.

In general, we agree with the reviewer on this point and have added conditions in which G3 fusion constructs are pre-loaded with soluble N-acetyllactosamine prior to adding them to protein-coated surfaces (Fig. 3). Note, that N-acetyllactosamine was used following recommendations made by reviewer #3, which are discussed below. We did not include controls

in which WT-G3 or G3 fusion constructs were pre-loaded with lactose before applying them to the column. In our reported experiments, the proteins were eluted by flushing the column with an increasing gradient of soluble lactose, which we feel is a suitable competitive binding control to demonstrate binding specificity. Likewise, the reproducible elution of WT-G3, as well as NL-G3, GFP-G3, and ChABC-G3, across each of our experiments further support the conclusion that this interaction is specific.

4. Suppl. Fig 1: Why not run WT G3 on this gel for comparison?

We appreciate this recommendation and have included new gels that allow for comparison of WT-G3 against the various reported G3 fusion constructs (Fig. S1). As reported above in response to reviewer #1, we have also expanded the molecular weight range of the gel to allow evaluation of protein purity.

5. Figure 4 and methods: the luciferase substrate emits in the blue spectrum - not ideal for in vivo monitoring due to overlap with other biological components.

We agree with the reviewer that some luminescence emitted by Nanoluc in vivo may be absorbed by surrounding tissues, especially in the case when G3 fusions are injected into intramuscular sites. We have added text and a reference on page 18 of the revised manuscript that addresses this point. However, previous reports demonstrate that Nanoluc is suitable for superficial in vivo imaging, such as in the mammary fat pad (Stacer, *AC Mol Imaging*, 2013), which supports our position that Nanoluc is suitable for in vivo imaging in the subcutaneous space. We also feel that Nanoluc provides a notable advantage over other methods, such as imaging proteins labeled with near-IR dyes, because it provides a direct measure of in vivo enzyme activity. In contrast, conjugation of near-IR dye would only provide half-life for any portion of the protein to which it is conjugated, and therefore may not accurately reflect half-life of enzyme activity.

6. Figure d, e, f: how reliable are these optical emission values (y-axis, $\sim 10^9$) for quantitative comparison since the enzyme and luciferase substrate injections are highly variable at tissue sites and the substrate has some PBS solubility issues as well? How are these y-axis time-zero Photon flux values so tight and similar for Figures 4d, e, f across all animals and tissue injection sites when they are usually highly variable?

We appreciate this point by the reviewer. The y-axis time zero photon flux values have reasonable variability, which may not have been apparent in the initial version of the manuscript because the data were presented on a log-linear plot. We apologize for this lack of clarity. To address this, we have included the individual photon flux data points at time zero for each animal within each group in the revised supplementary materials (Fig. S25). We note that in general the

WT-NL group demonstrates lower signal intensity at $t = 0$, which may reflect the fact that the protein is more rapidly flushed out of the injection site via convection associated with injecting a significant volume of buffer into a small tissue site because it cannot bind the tissue. In contrast, the mean photon flux values for NL-G3 and NL-TT-G3 are not significantly different from one another at any tissue site at $t = 0$. Thus, although some injection site variability for the enzyme and substrate is likely, we are confident that the optical emission values are suitable for comparing the duration of retention and half-life of NL-G3 and NL-TT-G3 at these different tissue sites. Likewise, any PBS solubility issues for the substrate are likely similar across our different experimental groups.

7. Page 17, endotoxin: LPS values are not reported anywhere?

Methods used for endotoxin removal and determination of endotoxin level were included in the initial version of the manuscript, under the heading, “**Cloning, expression, and purification of recombinant proteins**”. Under all circumstances, proteins were only used for in vivo studies when the endotoxin contamination level was below 1.0 EU/mL, which is an accepted threshold for preclinical evaluation of proteins (Mayala & Singh *J. Pharm. Sci.* 2008).

8. Page 18: Nano-Glo substrate, furimazine, excitation and emission wavelengths should be reported here despite use of an "open filter".

Because this assay measures luminescence, no excitation is necessary. On our machine, a separate light path without monochromators carries the emitted light to a dedicated photomultiplier tube. For “open filter” the optimal emission wavelength is between 360-630 nm. For all experiments, the integration time was held constant at 500 ms. These details have been included in the methods section of the revised manuscript.

9. Page 18: “Catalytic activity of 4 nM WT-NL, NL-G3, or NL-TT-G3 in wild-type female C57BL/6 mouse serum...”: complete undiluted serum?

We apologize for this oversight, and appreciate the reviewer bringing it to our attention. Serum was diluted 2-fold with 1x PBS containing 8 nM WT-NL, NL-G3, or NL-TT-G3 and then diluted another 2-fold with furimazine in PBS, giving a final concentration of 25% serum. These details have been included in the methods section of the revised manuscript.

10. Page 19: “.. we instead used the molar concentration of glycoprotein/proteoglycan adsorbed onto the plate as the total ligand concentration...” how was this adsorbed amount determined on surfaces?

As also noted by reviewer #3, the assumptions made regarding ligand concentration on the plate were imprecise. We have removed this text and the associated data from the manuscript, and have replaced it with data from saturating binding experiments that more accurately represent the relative affinity of G3 fusion constructs for adsorbed glycoproteins (Fig. 3e).

11. P. 20: “black, glass-bottom 96-well plates....”: since typical glass is not able to transmit UV wavelengths effectively below 360nm, was this UV-grade optical glass or quartz as required for such assays? What is its optical transmissivity at 280nm?

We greatly appreciate this recommendation by the reviewer. Upon further analysis, we determined that the optical transmissivity of our black, glass-bottom plates at 280 nm was insufficient to provide a preferable signal-to-noise ratio. To address this, we have repeated these experiments using quartz cuvettes instead of glass-bottom plates. In addition to repeating characterization of lactose binding (Fig. S6), we have also added analysis of n-acetyllactosamine binding (Fig. 3f) in response to recommendations by reviewer #3.

12. P. 20: Trp fluorescence: please show sample excitation and emission vs wavelength curves in the supplemental information for before and after lactose binding and how these were used to calculate (delta)RFU as reported.

We appreciate this comment, and have added sample curves to the supplemental information for each protein construct before and after lactose and n-acetyllactosamine binding (Fig. S20-21). We have also added a control sample of Nanoluc plus lactose over the entire sample range, which demonstrates that lactose does not interact non-specifically with Nanoluc (Fig. S6). We have also compared the mean minimum delta RFU and mean maximum delta RFU for each Trp fluorescence quenching experiment, which demonstrates that these signals are significantly different from each other. These data are also included in the supplemental information of the revised manuscript (Fig. S22).

13. P. 20, last line: please provide the assay LOD, sensitivity, and delta in optical signal at 420nm for aggregation? What (delta amplitude) difference measured is actually significant for light scattering? Please show sample curves for this difference in the Supplemental figures.

We greatly appreciate this recommendation by the reviewer, as the follow-up work has significantly strengthened our understanding of these molecules. We have added new data to the supplemental materials (Fig. S23) that describes the limit of detection, sensitivity, and delta in optical signal quantitatively, as well as differences in turbidity qualitatively. Based on these data, we increased the concentration of ASF and range of G3 constructs used in these assays. Our new results (Fig. 4g, revised), demonstrate that NL-TT-G3 can crosslink ASF at higher protein concentrations, while NL-G3 cannot. However, the change in optical signal at 420 nm for NL-

TT-G3 plus ASF was lower than that for WT-G3 plus ASF. To further evaluate these relationships, we have included a new assay characterizing the kinetics of ASF crosslinking (Fig. S11). These data demonstrate that WT-G3 forms aggregates with ASF faster than NL-TT-G3. Furthermore, to complement these data, we included another assay in which ASF crosslinking by G3 constructs was characterized using dynamic light scattering (Fig. S10). These data again demonstrate that NL-TT-G3 and WT-G3 can crosslink ASF into aggregates, while NL-G3 cannot, and also suggest that aggregates formed via WT-G3 are larger than those formed via NL-TT-G3. Finally, we have added bright-field photomicrographs (Fig. S9), which demonstrate that NL-TT-G3 and WT-G3 can form aggregates with ASF at high G3 concentration but not at low G3 concentration. The manuscript text has also been edited to reflect these new observations and discuss how they may relate to observed changes in biological activity of galectin-3 on pages 15-16.

14. P. 21: "Cell-Titre Blue": this is not an apoptosis-specific assay - this is a general cell viability/toxicity assay. Apoptosis is not shown.

We agree with the reviewer on this point, and apologize for inaccurate usage of terminology. Per recommendations by reviewer #1, we have edited the text and y-axis of this figure (Fig. 4e in the revised manuscript) to indicate that this assay measures metabolic activity. Furthermore, we have added new data evaluating phosphatidylserine exposure via Annexin V, which is typically used as an early marker of apoptosis, and membrane permeability, as determined via propidium iodide, to the supplemental materials (Fig. S7-8). Unexpectedly, we observed that NL-TT-G3 can induce phosphatidylserine exposure on Jurkat T cells without inducing agglutination, propidium iodide permeability, metabolic activity loss, or IL-2 secretion. Previous reports (Stowell SR, *J Immunol* 2008) demonstrate that galectin-3 can induce phosphatidylserine exposure on leukemic T cells without inducing death, and we expect that a similar response may be involved here. We have added text discussing this point to the manuscript on page 13. Finally, because we did not explicitly evaluate activation of apoptosis signaling pathways, we have edited replaced the word "apoptosis" with "death" or "loss of viability", where appropriate, when discussing our data in the revised manuscript.

15. P. 21: since G3 is a known T-cell activator, relevant changes in cytokine production should also be shown.

We greatly appreciate this recommendation by the reviewer. We agree that while T cell death is a common assay for galectin-3 activity at high concentrations, it has been shown that galectin-3 can induce IL-2 expression by Jurkat T cells at low dose [Hsu DK *Am J Pathol* 1996]. In the revised manuscript we have included data from an experiment quantifying IL-2 secreted by Jurkat T cells treated with WT-G3, NL-G3, or NL-TT-G3 in the presence or absence of lactose as inhibitor (Fig. 4f). These data demonstrate that Jurkat T cells secrete significantly more IL-2 than untreated cells, as expected. In contrast, cells treated with an equivalent amount of NL-G3 or NL-TT-G3 do not secrete more IL-2 than untreated cells. Thus, these data demonstrate that

G3 fusion constructs do not share activity for inducing cytokine secretion by T cells with WT-G3. Text describing these observations and their relationship to observations related to Jurkat T cell death induced by WT-G3 or G3 fusion constructs has also been added to the manuscript on page 14.

16. P. 21: why different time points for cell surface binding assay versus cell viability assay?

We appreciate this comment by the reviewer. We initially designed these experiments based on the concept that galectin-3 can bind to cell surfaces quickly, while changes in phenotype or function of cells will typically have a delay. Thus, we evaluated binding after 1 h and changes in metabolic activity after 4 h, where the latter is consistent with methods reported previously (Stowell SR, J Immunol 2008). However, to address this critique, we have repeated binding and agglutination experiments so that they are evaluated at the same time as the cell viability assay (4h). These data are presented in figure 4a-d of the revised manuscript.

17. P. 24: ELISA assay for anti-G3 antibodies in murine plasma: what's the positive control to show that this ELISA can detect anything specific? The assay should used spiked controls for a standard curve using known G3 additions to murine plasma. There's no proof provided that this assay can work at all. Please show an ELISA standard curve in the supplemental figures. Only two injections of antigen? How is this justified?

We apologize for any confusion here. This experiment was not designed to evaluate anti-G3 antibodies in murine serum. Rather, because of the relatively high sequence homology between human and mouse galectin-3 (~80%), our expectation was that any antibodies raised would likely be against Nanoluc. Thus, our positive control consisted of Nanoluc emulsified in an immunostimulatory adjuvant (TiterMax). Our data demonstrate that mice receiving NL in TiterMax do elicit anti-NL antibodies, while mice that received NL, NL-G3, or NL-TT-G3 without adjuvant did not (Fig. S14). Please note, “spiked controls for a standard curve using known G3 additions to murine plasma” are not a relevant control here, since the assay is evaluating presence of IgG in serum, not G3. Likewise, an “ELISA standard curve” is not relevant here because there is no standard to compare against. The relevant comparators are mice receiving protein plus adjuvant (positive control) and naïve cohorts (negative control).

Related to the last question, “*Only two injections of antigen? How is this justified*”, we and reviewer #1 agree this is a difficult experiment to design and conduct because there is not a single, all-encompassing vaccination schedule, and that outcomes of such an experiment will likely depend on the protein that is conjugated to G3, the disease state of the host, number of injections, frequency of injections, site of injections, and dose of injections. As discussed in response to reviewer #1, we initially chose two doses one month apart because this is sufficient to elicit high anti-protein serum IgG titers when the protein is administered with an adjuvant, as demonstrated with our positive control data for both nanoluciferase and TT-GFP emulsified in TiterMax. We believe that this experimental design allows us to ask the fundamental question,

‘does prolonging local retention increase host generation of anti-protein antibodies reactive against G3-fusion proteins’, to which our data suggest that the answer is no. However, we fully recognize that this schedule may not be therapeutically relevant. Rather, we envision that proteins will need to be delivered more frequently and possibly at higher doses to fully evaluate immunogenicity.

To address this critique, we have (1) edited the language in the text to more clearly outline our experimental design and motivation for these studies, (2) moved these data to the supplemental materials to de-emphasize their significance in this study (now Fig. S14), and (3) removed text in the abstract and conclusion attesting to the lack of immunogenicity of these fusion proteins and assemblies.

Reviewer #3

1) The concept of enzyme/galectin fusion is not new and has been formerly reported by Qasba et al. (Biochem. Biophys. Res. Commun. 2010, 394, 679-684) and Chammas et al. (Histochem. Cytochem. 2007, 55, 1015-1026), both references are not mentioned in the manuscript.

We appreciate this recommendation by the reviewer, which was also raised in other contexts by reviewers #1 and 2. To comply with editorial guidelines, we had to make decisions regarding references to include and exclude, which unfortunately led us to neglect some previous reports that are relevant to our work. In the initial manuscript, we had cited two previous reports of enzyme-galectin fusions (Kupper, *CE Curr. Pharm. Des.* 2013 & Bocker, *S. Glycobiology* 2017) because they provided relevant comparisons to galectin-3 binding data included in our report. However, we absolutely agree that these additional references represent notable contributions to the area of galectin-enzyme fusions. To address this critique, we have included discussion of the Qasba and Chammas references, alongside those by Kupper and Bocker, in the conclusion of our report (pgs 21-22), where we use them to frame the advances provided by our platform. Specifically, while galectin-enzyme fusions have been reported before, our work provides the first demonstration that these fusions are applicable for locally retaining enzyme activity *in vivo*. Additionally, our work provides the first demonstration that assembly of galectin-3 fusions into supramolecular architectures can be used to tune carbohydrate-binding affinity, and, in turn, *in vivo* half-life of enzyme-galectin fusions. Thus, while galectin-enzyme fusions are not entirely new, our report significantly advances the state-of-the-art for their use in biomedical applications.

2) The authors do not explain why they chose Gal-3 as lectin component – other lectins are also useful to prolong enzyme activities in tissues. The impact of these protein constructs may also lie in the fact that Gal-3 is a biomarker for tumor progression and angiogenesis.

We appreciate this comment by the reviewer. While we agree that many lectins could be effective for prolonging enzyme activity in tissues, we feel that Gal-3 is a particularly good choice for various reasons, some of which are outlined in detail in the introduction. First, Gal-3 is a human protein which makes it more applicable than non-human lectins in the context of drug delivery. Second, it is relatively small, lacks disulfide bonds, and does not require glycosylation, which makes it a good candidate for cost effective recombinant expression. Third, among galectins, Gal-3 demonstrates more promiscuous binding because it recognizes both glycosaminoglycans and beta-galactosides, suggesting it will have significantly more binding sites within tissues. Fourth, the intrinsically disordered N-terminal domain of Gal-3 provided a good candidate linker domain to ensure that the carbohydrate-recognition domain is sufficiently far from the TriggerTrimer or enzyme domains. Finally, we agree that Gal-3 fusions may provide advantages for retaining enzymes at sites of angiogenesis or within tumors given the role of Gal-3 as a biomarker within these sites. However, we feel that evaluation of Gal-3 fusions and nanoassemblies in these contexts is beyond the scope of this report. Furthermore, we expect that swapping the Gal-3 domain for other galectins or their subdomains may provide additional opportunities for local retention that are not afforded by Gal-3, studies of which are again beyond the scope of this report.

3) Gal-3 is very well characterized in terms of their glycan ligands on glycoproteins. This is not sufficiently addressed in the manuscript and parts of glycan binding characterization suffers from some weak points making subsequent conclusions questionable.

We agree with the reviewer on this point and have added a significant number of new experiments, as well as new GFP-G3 and GFP-TT-G3 fusion proteins, to the manuscript to address this critique. Specifically:

Page 8, first paragraph:

The authors' statement: "Significantly more enzyme bound to adsorbed glycoconjugates when formulated as NL-TT-G3 nanoassemblies versus monomeric NL-G3 fusion proteins on a G3 molar basis (Fig. 1i, j, Fig. 2a, Supplementary Fig. 3), demonstrating that NL-TT-G3 had higher apparent binding affinity for surface adsorbed glycoconjugates than NL-G3. We estimated NL-G3 and NL-TT-G3 relative binding affinity for different glycoconjugates by assuming that the carbohydrate concentration on the surface was equal to the glycoconjugate concentration coated onto the plate."

(a) Higher amounts of adsorbed enzyme activity does not necessarily indicate higher lectin affinity. Theoretically, binding of only one Gal-3 moiety of NL-TT-G3 on the adsorbed glycoprotein would result in three-fold higher enzyme activity. Enzyme activity read-out is not suitable to assess lectin affinity characteristics.

We agree that enzyme activity does not necessarily indicate higher lectin binding affinity. However, the activity of nanoluc is linear as a function of concentration over the range studied, as shown in figure 2 and the supplemental materials (Fig. S2). Thus, enzyme activity is an accurate measure of the amount of nanoluc/G3 bound. Although we accounted for differences in the number of NL molecules in the monomer and nanoassembly by using equimolar amounts of NL, we agree that one bound NL-TT-G3 would provide 3-fold higher enzyme activity than one bound NL-G3. In general, though, the amount of NL-TT-G3 bound is more than three times higher than the amount of NL-G3 bound. Nonetheless, to alleviate concerns about using enzyme activity as a read-out for extent of binding, all binding experiments have been repeated with GFP-G3 and GFP-TT-G3 fusions. The results with GFP support the conclusions arrived at with nanoluc constructs. We have also softened language in the text when discussing data in figures 3a-c (page 10) to indicate that increased extent of binding “may” indicate higher affinity of the trimeric nanossemblies for glycoconjugates when compared to monomeric fusions.

(b) What is meant with [NL] added in Fig. 2a (x-axis). Is it a normalized value in terms of NL concentration of the constructs (NL-TT-G3 with three NLs and NL-G3 with one NL)?

We apologize for any confusion here. Yes, in all binding experiments reported in figure 3 of the revised manuscript, the concentration of NL or GFP, which is equivalent to the concentration of Gal-3, was held constant across groups. Thus, the concentration of fusion proteins in NL-G3 or GFP-G3 samples is three-fold higher than the concentration NL-TT-G3 or GFP-TT-G3 nanoassemblies. This is why [NL] or [GFP] is reported on the x-axis.

(c) In Fig 2a binding sites of laminin seem to be already saturated at 1 nM NL. Binding curves with binding signal over concentration of fusion construct with readouts directly associated with Gal-3 binding (anti-Gal-3 antibodies, anti-HisTag) are more suitable for demonstrating binding efficiency.

We appreciate these comments and agree with the reviewer on these points. To address this critique, we have added new data in which binding was evaluated with GFP-G3 and GFP-TT-G3 fusions (Fig. 3). Because the number of molecules of Gal-3 is equivalent to the number of GFP molecules in these constructs, they provide a direct read-out of the number of bound Gal-3 molecules. Note, the sensitivity of GFP is orders of magnitude lower than nanoluciferase, and therefore higher concentrations of GFP fusions were required for these binding studies. To further address these critiques, we have also added saturation binding data in which the extent of GFP-G3 or GFP-TT-G3 binding to adsorbed asialofetuin or laminin is compared as a function of GFP concentration (Fig. 3e). Finally, we have performed Scatchard analysis of these data (Fig.

S4), which suggests that GFP-G3 binds non-cooperatively to adsorbed glycoproteins, as expected, whereas GFP-TT-G3 binds with positive cooperativity. From these data we were able to extract dissociation constants for GFP-G3 binding to asialofetuin and laminin, however the complex binding of GFP-TT-G3 to these glycoproteins precluded accurate estimation of K_D .

(d) The assumption that glycan concentration equals the concentration of adsorbed glycoproteins is not correct. Glycoconjugates are known to be only partially modified with glycans. Therefore, the presented determination and calculation of dissociation constant (Fig. 2b) is highly questionable.

We agree with the reviewer and have removed all instances of this assumption, as well as the data derived from it, from the manuscript.

(e) Instead, competitive binding assays giving IC50 values for competitive glycan ligands (lactose, LacNAc) should be applied. Binding strength of the Gal-3 fusion proteins should be then compared by IC50 values derived from competitive binding curves with soluble sugars/glycans with LacNAc (Galbeta1-4GlcNAc) as competitive soluble glycan ligand.

This was an excellent suggestion. To assess specificity of G3 fusion protein and nanoassembly binding to adsorbed glycoproteins, we have introduced new controls in which soluble LacNAc is used as the competitive ligand (Fig. 3a-b). We have also added competitive binding curves using LacNAc as the ligand (Fig. 3d), which suggest that the IC50 for LacNAc to inhibit NL-TT-G3 binding to ASF is significantly higher than that required to inhibit NL-G3 binding.

(f) Page 7, Fig 2c:

The experiment should exclude whether Gal-3 constructs are bound to GAGs by ionic interaction. A suitable control should be included. What are the pI values of the Gal-3 constructs?

We agree with the reviewer that non-specific interactions could be involved in GAG binding to Gal-3 constructs. However, the NL-G3 construct has a net charge of -6, while NL-TT-G3 has a net charge of -4, and therefore are not likely to bind anionic GAGs non-specifically. Additionally, competitive binding data using LacNAc demonstrate binding of NL-TT-G3 or GFP-TT-G3 to aggrecan is specific (Fig. 3a-b). Thus, we do not expect that Gal-3 constructs are binding to GAGs non-specifically. Nonetheless, because a rigorous control is difficult to establish for the GAG/laminin competitive binding studies reported in figure 2c in the initial submission, we have moved these data to the supplemental materials of the revised manuscript to de-emphasize their significance (Fig. S5). We have also added new text to the manuscript on page 12 explaining why we do not expect significant non-specific interactions between Gal-3 and GAGs.

(g) Page 9 and Fig.2d:

The conclusions are solely based on lactose (Fig 2d), which is not a constituent of glycans presented on the glycoconjugates used in this study. From this point of view, binding experiments with free lactose are not meaningful. It is well known from literature that LacNAc and poly-LacNAc. e.g. on laminin, are the glycans recognized by Gal-3. Conclusions from lactose binding studies of Gal-3 can therefore not be valid for glycoproteins, which vary in their content of LacNAc and poly-LacNAc.

We appreciate this important suggestion and have added new experiments to the revised manuscript that compare binding of LacNAc to Gal-3 constructs using tryptophan fluorescence quenching (Fig. 3f) and have moved the lactose binding data to the supplemental materials (Fig. S6). Our data do suggest that NL-G3 and NL-TT-G3 have higher affinity for LacNAc than wild-type Gal-3, although the differences are not as significant as what we observed previously for lactose. We have edited the text in the manuscript on pages 12-13 to discuss these relationships in greater detail.

(h) Apparent binding constants could be derived from binding experiments with immobilized asialofetuin (ASF), which is standard glycoprotein for galectin binding experiments with known N-glycan composition.

This was an excellent suggestion and we agree with the reviewer on this point. We have added these experiments to the revised manuscript in figure 3e.

(i) The following points should be addressed in the 'Methods' part:

- *What is the origin of the lactose affinity chromatography column?*

The resin was purchased from Sigma-Aldrich. The column was packed in-house according to methods provided by the supplier of the column unit (GE). Text describing these details has been added to the Methods section under the heading, "**Carbohydrate binding assays**".

- *Show calibration curves for NL activity over NL concentration (supplementary data).*

We have included these calibration curves, as well as calibration curves for GFP constructs to the supplemental materials (Fig. 2 and S2). We noted that nanoluciferase standard curves, while always linear, demonstrated variability in the magnitude of the RLU detected. Thus, a standard curve was produced for each individual nanoluciferase binding experiment to avoid analysis error between different experiments.

- *Carbohydrate bind assays: The calculation is highly questionable. The assumption that glycan concentration equals the glycoconjugate concentration is not valid. Glycans are only part of the glycoconjugates. The ligand concentration is much smaller than assumed.*

We agree, as does reviewer #2. This statement has been removed from the methods.

Reviewers' Comments:

Reviewer #1:

Remarks to the Author:

The authors have responded well to the comments of all reviewers. While there are immobilization strategies that use glycans and other ECM motifs, the combination of this with the trimerization is a novel design and gives benefit over current systems. In addition to therapeutics, there will be interest in this work from the protein engineering and assembly fields, as well as from those who study signal transduction in the ECM. I recommend this be accepted for publication.

Reviewer #2:

Remarks to the Author:

Many revisions have been made in response to prior critiques. These respond directly with technical accuracy and clarity expected. Some 30 new supplemental figures have been added to improve the message and details. Text is revised. Overall, the manuscript is highly improved and satisfies my prior concerns.

Reviewer #3:

Remarks to the Author:

The revised manuscript addresses all critical points. The additional experiments support the overall conclusions of the authors.

There is only one minor point:

The legend description for Figure 3 misses the assignment 'f' in the last two lines.